# Breaking the Echo Chamber: A Dynamic Ensemble Pruning Perspective on MoE

**Xinlai Kang** [* 1 2]  **Dunyao Xue** [* 1]  **Zhengbo Wang** [1]  **Chengshuo Du** [1]  **Xinghao Chen** [2]  **Hang Zhou** [3]
**Hanting Chen** [2]  **Cheng Meng** [4]

## Abstract

We introduce Mahalanobis-Pruned Mixture-of-Experts (MP-MoE), a novel routing framework that approaches expert selection from the perspective of ensemble pruning. Existing Mixture-of-Experts (MoE) routing strategies often suffer from representation collapse due to greedy top-k selection mechanisms or rely on complex auxiliary regularization terms that may compromise model performance. To address these issues, we formulate routing as a diversity-aware subset selection problem and optimize a Mahalanobis-distance-based objective that explicitly enhances expert diversity. Specifically, we demonstrate that the expert co-occurrence matrix effectively captures inter-expert correlations, allowing us to efficiently model the covariance structure required for distance computation without accessing expert parameters. Furthermore, we devise a greedy strategy for the routing mechanism, backed by theoretical approximation guarantees, rendering it a plug-and-play module with negligible overhead. MP-MoE increases wall-clock training time by approximately 3%, while incurring no additional latency at inference time. Extensive experiments demonstrate that during the pre-training of the large language model, our method consistently outperforms the baseline by 1-3 percentage points across a broad range of benchmarks.

[1] Institute of Statistics and Big Data, Renmin University of China, Beijing, China [2] Huawei Foundation Model Lab, Beijing, China [3] TJUNLP Lab, School of Computer Science and Technology, Tianjin University, Tianjin, China [4] Center for Applied Statistics, Institute of Statistics and Big Data, Renmin University of China, Beijing, China. Correspondence to: Cheng Meng <chengmeng@ruc.edu.cn>, Hanting Chen <chenhanting@huawei.com>.

*Proceedings of the 43rd International Conference on Machine Learning*, Seoul, South Korea. PMLR 306, 2026. Copyright 2026 by the author(s).

## 1. Introduction

Large Language Models (LLMs) (Radford et al., 2019; Brown et al., 2020) have demonstrated remarkable generalization capabilities across a wide range of fields, including natural language processing (Touvron et al., 2023; Chowdhery et al., 2023) and visual representation learning (Dosovitskiy et al., 2021; He et al., 2022; Jia et al., 2021; Radford et al., 2021). However, recent studies on scaling laws suggest that optimizing performance requires scaling model size and training data simultaneously (Fang et al., 2026b), which causes the inference cost of dense models to grow rapidly, thereby hindering practical deployment. To address this efficiency bottleneck, Mixture-of-Experts (MoE) (Shazeer et al., 2017; Fedus et al., 2022) architectures offer a promising alternative by decoupling model capacity from computational overhead. Unlike dense Transformers (Vaswani et al., 2017) that process every input with all parameters, MoE models utilize a routing mechanism to activate only a subset of specialized experts for each token. This mechanism enables models to scale to massive sizes with enhanced performance while maintaining a manageable computational budget and accelerating inference.

Despite the strong capacity of MoE, training can be hindered by expert collapse, which results in imbalanced routing and unused experts. Load-balancing methods (Lepikhin et al., 2021; Roller et al., 2021; Zhou et al., 2022) mitigate this issue via auxiliary objectives that encourage uniform expert utilization. However, while these methods effectively prevent expert idling, they inadvertently force disparate experts to process highly overlapping token distributions (Fang et al., 2026a). This homogenization of inputs leads to representation collapse (Chi et al., 2022), effectively trapping experts in an *'echo chamber'* where they fail to develop distinct specializations. While various strategies have been proposed to counteract representation collapse, they frequently entail significant trade-offs. StableMoE (Dai et al., 2022) decouples the training of routers and experts into two stages. However, this separation disrupts the co-evolution of the routing policy and expert capabilities. Similarly, SMoE-Dropout (Chen et al., 2023) and HyperRouter (Do et al., 2023) mitigate collapse by freezing randomly initialized routers, yet this renders the policy non-learnable, preventing adaptation to data distributions and limiting performance upper bounds.

*Table 1.* Comparison of different MoE routing mechanisms.

| Method | Routing Policy | Explicit Diversity[1] | Trainable Router | Aux. Free[2] |
|---|---|---|---|---|
| SMoE | Top-K | ✗ | ✓ | ✓ |
| StableMoE | 2-Stage | ✗ | ✗ | ✓ |
| SMoE-Drop | Fixed | ✗ | ✗ | ✓ |
| XMoE | Top-K | ✗ | ✓ | ✓ |
| HyperRouter | Fixed | ✗ | ✗ | ✓ |
| SimSMoE | Top-K | ✓ | ✓ | ✗ |
| Guo et al. (2025) | Top-K | ✓ | ✓ | ✗ |
| **MP-MoE** (Ours) | **Ensemble** | ✓ | ✓ | ✓ |

[1] **Explicit Diversity**: Explicitly penalizes inter-expert similarity.
[2] **Aux. Free**: Without additional auxiliary loss.

Other approaches like SimSMoE (Do et al., 2025) and CompeteSMoE (Pham et al., 2024) utilize expert embeddings to enhance diversity, but this typically necessitates the full activation of experts. Consequently, they require complex architectural workarounds to mitigate computational overhead, often resulting in information underutilization. Furthermore, methods relying on auxiliary regularization, such as Guo et al. (2025), must strictly balance the regularization weight against the primary objective, which introduces optimization conflicts that can degrade model performance. More recently, GatePro (Zheng et al., 2025) encourages diversity via a competition mechanism based on pairwise similarity of gating weights. However, gating weights lack a direct correlation with functional expert similarity, and the hard suppression is hyperparameter-sensitive, failing to capture redundancy beyond paired experts. These limitations motivate our key question:

> *How to design a simple and efficient framework to break the "echo chamber" among experts and foster collective expert diversity without sacrificing model performance?*

In this work, we address this question from the perspective of ensemble pruning, a paradigm that entails selecting a fixed subset of learners from a larger pool to optimize predictive outcomes. This paradigm closely mirrors the routing mechanism inherent in MoE architectures. Extensive research in ensemble learning establishes that superior generalization is achieved not merely by aggregating high-accuracy learners, but by balancing individual performance with collective diversity (Krogh & Vedelsby, 1994; Opitz & Maclin, 1999). Consequently, a naive selection strategy that solely prioritizes the highest-scoring learners is insufficient for attaining optimal group performance, as it often introduces redundancy among the selected members (Zhou et al., 2002; Li et al., 2012). We argue that MoE architectures suffer from similar limitations: the standard

Top-K routing, which greedily selects experts based on magnitude, fails to account for expert correlations. To overcome this, we leverage a key empirical insight: the expert co-occurrence matrix effectively captures intrinsic inter-expert correlations. Specifically, we examine the correspondence between expert similarity based on output embeddings and that based on activation patterns. The results demonstrate a high correlation and consistent clustering structures between the two, confirming that the co-occurrence matrix faithfully mirrors output-based similarity and serves as a reliable proxy for expert redundancy. Building on this, we introduce MP-MoE, a routing framework that incorporates these correlations to reformulate expert selection as a subset maximization problem driven by the Mahalanobis distance. By approximately solving this diversity-aware objective using an efficient greedy procedure, MP-MoE explicitly promotes the selection of functionally complementary experts during pre-training, thereby maximizing collective diversity without incurring significant computational overhead.

Our contributions are summarized as follows:

- We propose a novel routing framework grounded in ensemble pruning, which leverages the Mahalanobis distance to maximize the collective diversity of selected experts during pre-training.

- We first reveal the intrinsic link between expert co-occurrence patterns and their representational similarity, incorporating this insight into the routing mechanism to enhance expert diversity.

- We devise an efficient greedy algorithm for the routing mechanism and provide theoretical guarantees regarding its approximation to the optimal solution.

- We conduct extensive experiments to demonstrate the strong learning capability of MP-MoE. Our method increases wall-clock pre-training time by approximately 3%, while introducing no additional inference-time latency. Across a broad range of benchmarks, MP-MoE consistently outperforms the baseline by 1–3 percentage points.

## 2. Motivation

### 2.1. Background of Mixture of Experts

Mixture-of-Experts (MoE) models have become a foundational module in modern neural architectures, significantly reducing the active parameter count during inference through sparse routing mechanisms. Following (Fedus et al., 2022), we implement MoE specifically within the fully connected layers.

In the most widely used MoE architecture, each token representation $h \in \mathbb{R}^D$ selects $k$ experts via a softmax-based

router:

$$\text{MoE}(h) := \sum_{r=1}^{E} \text{Gate}_r(h) \cdot f_r^{\text{FFN}}(h),$$

$$\text{Gate}_r(h) := \text{top}_k(\text{softmax}(Wh + \epsilon))\,[r], \quad \forall r \in [E],$$

where the gating coefficients $\{\text{Gate}_r(h)\}_{r=1}^{E}$ linearly combine the outputs of the $E$ experts $\{f_r^{\text{FFN}}(h)\}_{r=1}^{E}$. The operator $\text{top}_k : \mathbb{R}^E \to \mathbb{R}^E$ enforces sparsity by setting all but the $k$ largest values to zero ($k \ll E$). The router parameters $W \in \mathbb{R}^{E \times D}$ are trained jointly with the other network parameters. To improve the numerical stability of the router, a noise vector $\epsilon \sim \mathcal{N}(0, \sigma^2 I)$ is injected into the logits $Wh$, where $\sigma$ controls the noise strength. Furthermore, auxiliary load-balancing objectives are typically incorporated during pre-training to encourage a uniform routing of tokens across all experts.

## 2.2. Connection with Ensemble Pruning

Ensemble pruning aims to select a subset of members that maintains or enhances the overall performance compared to the complete ensemble. Error analysis (Breiman, 2001) suggests that ensemble performance can be characterized by a trade-off between individual accuracy and pairwise diversity. Building on this idea, Zhang et al. (2006) cast ensemble pruning as a quadratic integer program by constructing an error co-occurrence matrix on a validation set. Let $P$ be the 0/1 error indicator matrix and define $G = P^\top P$, where $G_{ii}$ counts the errors of model $i$ and $G_{ij}$ counts the common errors of models $i$ and $j$. After normalization, one obtains a matrix $\tilde{G}$ that jointly encodes accuracy (diagonal) and redundancy (off-diagonal), leading to the following subset selection objective:

$$\min_{\mathbf{z}} \; \mathbf{z}^\top \tilde{G} \mathbf{z} \qquad \text{s.t.} \quad \sum_i z_i = K, \quad z_i \in \{0,1\}. \quad (1)$$

This quadratic form selects a size-$K$ sub-ensemble by penalizing both individual error and pairwise overlap. Notably, selecting a subset of base learners here is closely related to expert selection in MoE: both aim to activate only a small subset of candidates for each input while maintaining overall performance.

A key requirement in ensemble pruning is to simultaneously encourage accuracy and diversity among the selected learners, which aligns well with our goal of promoting expert diversity, i.e., reducing redundant expert activations and encouraging differentiated behaviors across experts.

## 2.3. Generalizing Ensemble Pruning to MoE via Mahalanobis Distance

The quadratic subset-selection objective in Eq. 1 utilizes second-order statistics via $\tilde{G}$ to penalize learners that 'fail

together,' thereby promoting a subset that balances accuracy with diversity. Motivated by this, we aim to extend this ensemble pruning paradigm to the MoE architecture. To achieve this generalization, we observe that for any random vector $\mathbf{z}$ and a symmetric positive definite matrix $\mathbf{S}$, this quadratic structure naturally corresponds to the Mahalanobis distance.

In modern machine learning, the Mahalanobis distance has been widely used to induce more discriminative and diversity-aware objectives. First, in metric learning, one learns a Mahalanobis metric to pull samples of the same class closer while pushing samples of different classes apart (Weinberger & Saul, 2009). Second, in out-of-distribution (OOD) detection, one can model class-conditional deep features as Gaussians and use a Mahalanobis-based confidence score to detect abnormal inputs (Lee et al., 2018). Moreover, in representation learning, Mahalanobis distance can also be used as a regularization term to improve classification and robustness (Bateni et al., 2020; Wan et al., 2018).

Formally, given two vectors $\mathbf{x}, \mathbf{y} \in \mathbb{R}^D$ and a symmetric positive definite matrix $\mathbf{S} \succ 0$, the Mahalanobis distance induced by $\mathbf{S}$ is defined as:

$$d_M(\mathbf{x}, \mathbf{y}; \mathbf{S}) := \sqrt{(\mathbf{x} - \mathbf{y})^\top \mathbf{S}^{-1} (\mathbf{x} - \mathbf{y})}. \quad (2)$$

In particular, when $\mathbf{y}$ is set to the origin (i.e., $\mathbf{y} = \mathbf{0}$) and $\mathbf{S}$ represents the covariance matrix $\mathbf{\Sigma}$, we obtain the Mahalanobis norm of $\mathbf{x}$:

$$d_M(\mathbf{x}, \mathbf{0}; \mathbf{\Sigma}) := \sqrt{\mathbf{x}^\top \mathbf{\Sigma}^{-1} \mathbf{x}}. \quad (3)$$

Equivalently, if $L^\top L = \mathbf{\Sigma}^{-1}$, then $d_M^2(\mathbf{x}, \boldsymbol{\mu}) = \|L(\mathbf{x} - \boldsymbol{\mu})\|_2^2$, which shows that the Mahalanobis distance corresponds to Euclidean distance after a whitening transform, and therefore downweights correlated directions (De Maesschalck et al., 2000).

These demonstrate that by explicitly accounting for second-order dependencies via $\mathbf{\Sigma}^{-1}$, the Mahalanobis distance not only normalizes scale discrepancies across different dimensions but also facilitates information separation along non-redundant directions. Such properties are intrinsically aligned with our objective of enhancing expert diversity.

# 3. Method: Mahalanobis-Pruned Mixture of Experts

In this section, we propose the Mahalanobis-Pruned Mixture of Experts (MP-MoE) model. Building upon the principle of ensemble pruning, MP-MoE seeks an optimal set of experts through an optimization objective defined in terms of the Mahalanobis distance for each layer, thereby encouraging greater diversity across experts. The main structure of MP-MOE is illustrated on the right side of Figure 1.

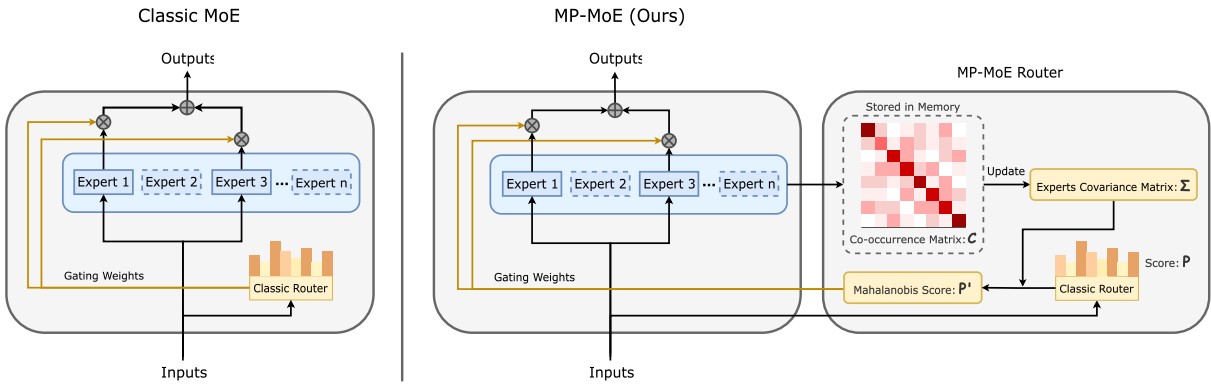

*Figure 1.* A comparison between the classic MoE architecture (left) and our MP-MoE architecture (right). MP-MoE extends the standard MoE formulation by modifying the router's scoring and selection procedure, thereby better preserving expert diversity during pre-training.

### 3.1. Mahalanobis Ensemble Routing

Inspired by ensemble pruning, we conceptualize each expert as a weak learner. To encourage specialization among these learners and maximize their collective representational power, we propose an enhanced routing mechanism. Departing from the traditional Top-$k$ strategy, we seek an optimal subset that maximizes the squared Mahalanobis distance of the experts from the origin.

Specifically, consider an MoE layer comprising $N$ experts $\mathcal{E} = \{E_1, \ldots, E_N\}$. For each token input $\mathbf{x}$, the gating network computes a set of routing scores. We aim to identify an optimal subset $\mathcal{E}_k^* \subset \mathcal{E}$ of size $k$ (where $k \ll N$) that maximizes the joint reliability and diversity of the selected experts. The optimization objective for the given token is formulated as:

$$\mathcal{E}_k^* = \underset{\mathcal{E}_k \subset \mathcal{E}, \, |\mathcal{E}_k|=k}{\arg \max} \left(\boldsymbol{\mu}_{\mathcal{E}_k}^{\top} \boldsymbol{\Sigma}_{\mathcal{E}_k}^{-1} \boldsymbol{\mu}_{\mathcal{E}_k}\right) \qquad (4)$$

where $\boldsymbol{\mu}_{\mathcal{E}_k}$ denotes the vector of routing logits corresponding to the selected $k$ experts for input $\mathbf{x}$, and $\boldsymbol{\Sigma}_{\mathcal{E}_k}^{-1}$ is the inverse of the inter-expert covariance matrix capturing the correlation structure among the chosen experts to penalize redundancy.

From the form of Equation 4, Mahalanobis Ensemble Routing can be regarded as an improvement over the classic MoE routing strategy. Classic MoE routing selects the $k$ highest-scoring entries from the score vector, which can be interpreted as choosing the subvector that attains the largest $\ell_1$-norm magnitude among the expert score vectors. In contrast, Mahalanobis Ensemble Routing selects the subvector that attains the largest Mahalanobis-norm magnitude among the expert score vectors.

### 3.2. Construction of Inter-Expert Covariance Matrix

A critical challenge in diversifying MoE lies in efficiently estimating the inter-expert correlation structure. Prior approaches, such as SimSMoE (Do et al., 2025) and CompeteSMoE (Pham et al., 2024), typically rely on expert embeddings to compute pairwise similarities or importance scores. However, these methods inherently necessitate the full activation of experts or the design of intricate architectures, thereby incurring prohibitive computational overhead and leading to information underutilization.

To address this challenge, we seek a metric that is both readily accessible and indicative of intrinsic expert similarities. We identify the expert co-occurrence matrix as an effective solution. As corroborated by Guo et al. (2025), experts exhibiting high co-occurrence frequencies are exposed to overlapping sets of tokens, indicating that they model similar data distributions and possess redundant functional domains. Based on this, we boldly posit that this shared exposure is the root cause of expert redundancy, as it forces experts to learn from similar contexts, ultimately leading them to converge towards homogenous representations during the training process.

To construct the expert covariance matrix using the selection patterns among experts, we model the selection status of the $i$-th expert for a given token as a Bernoulli random variable $I_i \in \{0, 1\}$. Under this probabilistic framework, the co-occurrence statistics can be naturally mapped to a covariance structure. Let $\mathbf{C} \in \mathbb{R}^{E \times E}$ denote the accumulated expert co-occurrence count matrix, where $\mathbf{C}_{ij}$ represents the number of tokens activating both expert $i$ and expert $j$. Let $\mathbf{u} = \text{diag}(\mathbf{C})$ be the vector of individual activation counts, and $N$ be the total token count. We estimate the covariance matrix $\boldsymbol{\Sigma}$ of the expert selection variables as:

$$\boldsymbol{\Sigma} = \frac{\mathbf{C}}{N} - \frac{\mathbf{u}\mathbf{u}^{\top}}{N^2} \qquad (5)$$

Here, the term $\mathbf{C}_{ij}/N$ serves as an unbiased estimator for

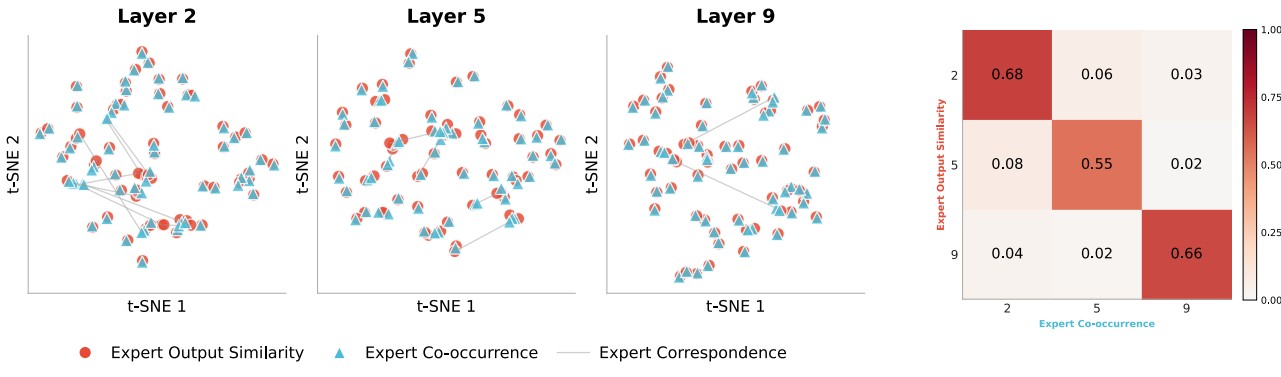

*(a)* t-SNE Embedding Comparison    *(b)* Layer-wise CKA Correlation

*Figure 2.* Validation of the co-occurrence matrix as a similarity proxy. (a) t-SNE embeddings of the expert output similarity (Red) versus the co-occurrence matrix (Blue), with diagonals replaced by the row-wise mean of off-diagonal elements to reduce self-correlation dominance. (b) Layer-wise Centered Kernel Alignment (CKA) results derived from the upper triangular components of both matrices. The spatial alignment in (a) and high CKA scores on the diagonal in (b) consistently demonstrate that expert co-occurrence serves as a robust indicator of expert similarity.

the joint expectation $\mathbb{E}[I_i I_j]$, while $(\mathbf{u}_i \mathbf{u}_j)/N^2$ approximates the product of marginal expectations $\mathbb{E}[I_i]\mathbb{E}[I_j]$. Thus, Eq. 5 effectively captures the pairwise covariance structure derived from routing behaviors.

To validate the correlation between this co-occurrence-derived covariance matrix and the expert similarity, we conduct a post-hoc analysis on a fully trained MoE model. For a specific MoE layer, we forward identical tokens to all experts to extract their output embeddings and compute the pairwise cosine similarity matrix. We then compare this representation-based similarity with the co-occurrence covariance matrix accumulated during inference. As visualized in Figure 2a, the t-SNE projections of both matrices exhibit remarkable structural alignment across different layers. To quantify this observation, we flattened and concatenated the upper triangular elements of both the expert output similarity matrix and the expert co-occurrence matrix to compute their Centered Kernel Alignment (CKA) similarities. The resulting coefficients are consistently around 0.6, indicating a strong statistical correlation between the two. This confirms that our co-occurrence-based metric effectively captures the intrinsic representational similarity among experts.

### 3.3. Optimization Strategy

Direct maximization of the objective in Eq. 4 poses a combinatorial challenge known to be NP-hard. To ensure computational tractability, we adopt a forward selection strategy. A naive implementation would require computing the matrix inverse $\mathbf{\Sigma}_{\mathcal{E}_k}^{-1}$ at every step, incurring a prohibitive cubic cost $\mathcal{O}(k^3)$. To mitigate this, we propose an efficient algorithm based on incremental Cholesky factorization (Golub & Van Loan, 2013). This approach allows us to update the in-

verse Cholesky factor $\mathbf{R}$ recursively, significantly reducing the computational complexity.

The selection criterion in Algorithm 1 can be interpreted through the lens of Orthogonal Matching Pursuit (Pati et al., 1993). Specifically, in each iteration, the term $(p_j - \boldsymbol{\alpha}_j^\top z)$ represents the residual score that is orthogonal to the currently selected experts. The term $r_j$ acts as a normalization factor based on the conditional variance. Consequently, the algorithm greedily selects the expert that offers the maximum marginal gain relative to the existing subset, effectively balancing the raw routing score with the redundancy penalty.

To theoretically validate the proposed greedy selection strategy, we invoke the framework established by Das & Kempe (2018). The key lies in quantifying how much the objective function deviates from submodularity. The following theorem establishes that the approximation gap between the greedy solution and the optimal solution is controlled by the submodularity ratio.

**Theorem 3.1** (Approximation Guarantees for Mahalanobis Ensemble Routing). *Let* $f(\mathcal{E}_k) = \boldsymbol{\mu}_{\mathcal{E}_k}\mathbf{\Sigma}_{\mathcal{E}_k}^{-1}\boldsymbol{\mu}_{\mathcal{E}_k}^T$ *and* $OPT = \max_{|S| \leq k} f(S)$ *, combining Lemma A.1 and Lemma A.2 in the appendix, the set of experts* $\mathcal{E}_k$ *selected by the Mahalanobis Ensemble Routing algorithm satisfies the following approximation guarantees:*

$$
\begin{aligned}
f(\mathcal{E}_k) &\geq (1 - e^{-\gamma_{\mathcal{E}_k,k}}) \cdot OPT \\
&\geq (1 - e^{-\lambda_{\min}(\mathbf{\Sigma},2k)}) \cdot OPT \\
&\geq (1 - e^{-\lambda_{\min}(\mathbf{\Sigma},k)}) \cdot \Theta\left(\left(\frac{1}{2}\right)^{1/\lambda_{\min}(\mathbf{\Sigma},k)}\right) OPT,
\end{aligned}
$$

*where* $\gamma_{U,k}$ *is the submodularity ratio of* $f$ *defined as the minimum ratio of the marginal gain of a set to the marginal*

---

**Algorithm 1** Greedy Algorithm for MoE Routing

---

1: **Input:** Scores of $E$ experts, $\boldsymbol{\mu} \in \mathbb{R}^E$, Experts covariance $\boldsymbol{\Sigma} \in \mathbb{R}^{E \times E}$; Number of selected Experts $k$.
2: **Initialize** $\mathcal{E}_k = \varnothing$
3: Select the first Expert
4: $k_1 = \arg\max_{i \in \{1, \dots, E\}} \frac{\mu_i^2}{\Sigma_{i,i}}$
5: $\mathcal{E}_k \leftarrow \{k_1\}$
6: $\mathbf{R} = (\boldsymbol{\Sigma}_{\mathcal{E}_k, \mathcal{E}_k}^{1/2})^{-1}$, $z \leftarrow \mathbf{R}\boldsymbol{\mu}_{\mathcal{E}_k}$
7: **for** $i = 2$ **to** $k$ **do**
8:    **for** $j \in \{1, \dots, E\} \setminus \mathcal{E}_k$ **do**
9:       Let $\boldsymbol{\beta}_j$ be the vector from $\boldsymbol{\Sigma}$ consisting of rows $\mathcal{E}_k$ and column $j$
10:       Let $b_j$ be the $j$th diagonal element of $\boldsymbol{\Sigma}$
11:       $\boldsymbol{\alpha}_j \leftarrow \mathbf{R}\boldsymbol{\beta}_j$
12:       $r_j \leftarrow (b_j - \boldsymbol{\alpha}_j^\top \boldsymbol{\alpha}_j)^{-1/2}$
13:       Compute score:
14:
$$c_j = \|z\|_2^2 + \left(r_j \left(\mu_j - \alpha_j^\top z\right)\right)^2$$
15:    **end for**
16:    Select $j_* = \arg\max_j c_j$
17:    $\boldsymbol{\gamma} \leftarrow -r_* \mathbf{R}^\top \boldsymbol{\alpha}_*$
18:    Update
19:
$$\mathbf{R}_{new} = \begin{bmatrix} \mathbf{R} & \mathbf{0} \\ \boldsymbol{\gamma}^\top & r_* \end{bmatrix} \qquad z_{new} = \begin{bmatrix} z \\ r_* \left(\mu_{j_*} - \alpha_*^\top z\right) \end{bmatrix}$$
20:
$$\mathbf{R} \leftarrow \mathbf{R}_{new} \quad z \leftarrow z_{new}$$
21:
$$\mathcal{E}_k \leftarrow \mathcal{E}_k \cup \{j_*\}$$
22: **end for**
23: **Output:** Set of selected Experts $\mathcal{E}_k$.

---

*gain of its individual elements:*

$$\gamma_{U,k} = \min_{L \subseteq U, A:|A| \leq k, L \cap A = \emptyset} \frac{\sum_{x \in A} \left(f(L \cup \{x\}) - f(L)\right)}{f(L \cup A) - f(L)}.$$

*and* $\lambda_{\min}(\boldsymbol{\Sigma}, k) \triangleq \min_{S:|S|=k} \lambda_{\min}(\boldsymbol{\Sigma}_S)$ *denotes the smallest eigenvalue among all $k \times k$ principal submatrices of $\boldsymbol{\Sigma}$.*

*Detailed proofs are provided in the Appendix A.*

This theorem provides a rigorous batch-wise approximation guarantee for our method. Crucially, the lower bound on $\lambda_{\min}$ is maximized when the selection probabilities $P_j$ are balanced across the batch. This theoretical insight implies that improved load balancing directly enhances the effectiveness of our greedy algorithm, motivating our practical choice to integrate traditional load balancing auxiliary

losses. Furthermore, as the model trains and expert diversity increases, the approximation bound tightens further, suggesting that the routing performance progressively improves throughout the training dynamics.

### 3.4. Training Framework of MP-MoE

Building upon the Mahalanobis Ensemble Routing strategy and the covariance matrix derived from expert co-occurrence, we present the specific training procedure for MP-MoE. To simplify the exposition, the process for a single-layer architecture is outlined in Algorithm 2, while Figure 1 illustrates the corresponding workflow.

---

**Algorithm 2** MP-MoE Training Framework

---

1: **Input:** Training data $\mathcal{D}$, Experts set $\mathcal{E}$, Selection size $k$.
2: **Initialize:** Global Co-occurrence Matrix $\mathbf{C} \leftarrow \mathbf{0}_{E \times E}$, Global Covariance Matrix $\boldsymbol{\Sigma} \leftarrow \mathbf{I}_{E \times E}$, Total token count $N_{total} \leftarrow 0$, Parameters $\theta$.
3: **while** training not finished **do**
4:    Sample batch $\mathbf{X} \in \mathbb{R}^{B \times S \times L}$ from $\mathcal{D}$
5:    Flatten $\mathbf{X}$ to tokens $\mathbf{T} \in \mathbb{R}^{N_{batch} \times L}$, where $N_{batch} = B \times S$
6:    $N_{total} \leftarrow N_{total} + N_{batch}$
7:    $\mathbf{M} \leftarrow \text{GatingNet}(\mathbf{T}; \theta)$
8:    $\mathbf{u} \leftarrow \text{diag}(\mathbf{C})$
9:    $\boldsymbol{\Sigma} \leftarrow \frac{\mathbf{C}}{N_{total}} - \frac{\mathbf{u}\mathbf{u}^\top}{N_{total}^2} + \epsilon\mathbf{I}$
10:    Initialize binary selection mask $\mathbf{S}_{batch} \in \{0,1\}^{N_{batch} \times E} \leftarrow \mathbf{0}$
11:    **for** $n = 1$ **to** $N_{batch}$ **do**
12:       $\boldsymbol{\mu} \leftarrow \mathbf{M}_n$
13:       $\mathcal{E}_k \leftarrow \text{Mahalanobis Ensemble Routing}(\boldsymbol{\mu}, \boldsymbol{\Sigma}, k)$
14:       Set $\mathbf{S}_{batch}[n, j] \leftarrow 1$ for all $j \in \mathcal{E}_k$
15:    **end for**
16:    Compute Loss $\mathcal{L}$ using $\mathbf{S}_{batch}$ and selected experts
17:    Update $\theta \leftarrow \text{Optimizer}(\theta, \nabla_\theta \mathcal{L})$
18:    $\mathbf{C} \leftarrow \mathbf{C} + \mathbf{S}_{batch}^\top \mathbf{S}_{batch}$
19: **end while**

---

Algorithm 2 demonstrates that MP-MoE necessitates only the tracking of expert co-occurrence statistics. Coupled with the efficient greedy algorithm from Section 3.3, our method achieves near-zero additional overhead, making it an effective plug-and-play solution for diverse MoE backbones.

## 4. Experiments

### 4.1. Experimental Setup

**Pre-training Data and Training Budget.** We conduct pre-training experiments on a 100B-token subset of the FineWeb-EDU corpus (Penedo et al., 2024). In investigating the impact of router behavior during the early stages of pre-training, we consider three training regimes

of increasing scale under different compute budgets: 1B, 5B, and 50B tokens. Our code is available at https://github.com/kxlkxl1999/MP-MoE.

**Benchmarks and Metrics.** We evaluate pre-trained checkpoints on a fixed suite of six standard benchmarks: MMLU(Hendrycks et al., 2021), BoolQ(Clark et al., 2019), HellaSwag(Zellers et al., 2019), BBH(Suzgun et al., 2023), ARC-Easy, and ARC-Challenge(Clark et al., 2018). We follow the canonical evaluation protocol for each benchmark and report the corresponding standard metrics, including accuracy for MMLU, BoolQ, HellaSwag, ARC-Easy, and ARC-Challenge, and exact match for BBH. Unless otherwise noted, evaluation is performed without any task-specific fine-tuning.

**Model Configurations.** Our primary baseline is OLMoE-1B-7B, a Mixture-of-Experts model with 7B total parameters and 1B active parameters per token at inference. We also include a smaller configuration, OLMoE-0.2B-1B, which has 1B total parameters with 0.2B active parameters, 8 transformer layers (versus 16), hidden size 768 (versus 2048), and 6 attention heads (versus 16). Both architectures use MoE layers with 64-out-of-8 expert routing. MP-MoE denotes the same backbone architecture as our proposed router modification. Aside from the routing mechanism, all transformer components are kept identical to ensure controlled comparisons.

**Training Details.** All models are trained from scratch on the same 100B-token subset. We run three training settings: (i) 1B tokens on OLMoE-1B-7B, (ii) 5B tokens on OLMoE-1B-7B, and (iii) 50B tokens on OLMoE-0.2B-1B. For MP-MoE, we employ an MP warmup schedule in which MP routing is disabled for the first 1% of optimization steps, reverting to standard MoE routing to improve training stability. We use AdamW with a linear warmup over 1% of steps followed by cosine descent, and apply gradient accumulation to reach the target effective batch size. During training, we log an expert co-occurrence matrix $C$ that records how frequently experts are selected together, and we update the expert correlation matrix $\sigma$ every 10 iterations to progressively renew the MP mechanism. All experiments are conducted on Ascend 910B NPUs using the same sequence length and batch-size configuration to ensure fairness.

**Evaluation Protocol.** We evaluate all models in a zero-shot or few-shot manner directly from pre-trained checkpoints, without additional fine-tuning. The shot settings used in Table 2 follow common practice for each benchmark. At inference time, MP-MoE uses the same standard softmax top-$k$ routing as the OLMoE baseline. Consequently, the proposed modification affects training only and introduces no additional inference-time overhead. Reported results are averaged over at least three runs when applicable.

**Complexity Analysis.** In a classic MoE layer, the router computes a score for each of the $E$ experts via a linear projection of the token hidden state, incurring a time complexity of $\mathcal{O}(Ed)$. It then selects the top-$k$ experts by sorting or partial selection over the resulting logits, which has worst-case complexity $\mathcal{O}(E \log E)$. This gating step requires $\mathcal{O}(E)$ space per token to store all expert scores.

MP-MoE preserves the same initial gating computation $\mathcal{O}(Ed)$ to produce a score vector $\boldsymbol{\mu} \in \mathbb{R}^E$, but replaces the top-$k$ operation with a greedy Mahalanobis-based routing procedure. Importantly, we incrementally maintain an inverse Cholesky factor rather than recomputing $\boldsymbol{\Sigma}_{S,S}^{-1}$ from scratch at each step, leading to an overall routing cost of approximately $\mathcal{O}(Ek^2)$ per token. Since $k \ll E$, this scaling is effectively linear in $E$ and is comparable to classic top-$k$ gating in practice. The per-token space complexity remains $\mathcal{O}(E)$ for storing logits and intermediate scores. The covariance matrix requires $\mathcal{O}(E^2)$ memory but is fixed and shared across tokens, and the greedy updates introduce only an additional $\mathcal{O}(k^2)$ scratch buffer for Cholesky maintenance.

Computationally, MP-MoE's routing introduces only a marginal overhead over standard top-$k$ gating. In a classic MoE layer with $E = 64$ experts and hidden dimension $d = 2048$, computing the gating scores $W \cdot h$ costs on the order of $E \times d \approx 1.3 \times 10^5$ FLOPs per token, and the top-8 selection step ($\mathrm{O}(E \log k)$) is negligible by comparison. MP-MoE adds $k = 8$ rounds of greedy expert selection on these scores, each involving vector–matrix operations with the precomputed $E \times E$ covariance matrix $\Sigma$ to evaluate marginal gains. By leveraging incremental Cholesky updates and avoiding a full $k \times k$ matrix inversion at every selection step, the routing procedure introduces only about $5 \times 10^4$ additional FLOPs per token for $E = 64$ and $k = 8$. This amounts to an approximately 2–3% increase in pre-training compute, which is small relative to the overall forward-pass cost. Consistent with this estimate, we observe an approximately 3% increase in wall-clock pre-training time in practice.

**Inference Efficiency.** MP-MoE uses the same standard top-8 routing as the baseline during inference. Therefore, throughput, latency, and memory footprint remain unchanged relative to OLMoE at the time of deployment. This property allows models trained with MP routing to be served with existing MoE inference implementations without modification, while benefiting from improved pre-training quality.

*Table 2.* Benchmark Comparison between Classic MoE and MP-MoE on OLMoE

| Benchmark (Metric) | Trained Tokens | MMLU (Acc.) | BoolQ (Acc.) | HellaSwag (Acc.) | BBH (EM) | ARC-Easy (Acc.) | ARC-Challenge (Acc.) | Avg. |
|---|---|---|---|---|---|---|---|---|
| Shots | - | 5-shot | 0-shot | 0-shot | 3-shot | 0-shot | 0-shot | - |
| OLMoE 1B-7B (Baseline) | 1 BT | 0.245 | 0.563 | 0.243 | 0.121 | 0.265 | 0.255 | 0.282 |
| OLMoE 1B-7B (MP-MoE) | 1 BT | **0.255** | **0.597** | **0.271** | **0.169** | **0.311** | **0.269** | **0.312** |
| OLMoE 1B-7B (Baseline) | 5 BT | 0.258 | 0.568 | 0.251 | 0.132 | 0.298 | 0.254 | 0.294 |
| OLMoE 1B-7B (MP-MoE) | 5 BT | **0.262** | **0.604** | **0.273** | **0.172** | **0.327** | **0.275** | **0.319** |
| OLMoE 0.2B-1B (Baseline) | 50 BT | 0.249 | 0.560 | **0.246** | 0.114 | 0.249 | 0.255 | 0.279 |
| OLMoE 0.2B-1B (MP-MoE) | 50 BT | **0.264** | **0.583** | 0.243 | **0.127** | **0.270** | **0.257** | **0.291** |

## 4.2. Results and Analysis

**Overall Performance.** Table 2 summarizes the performance of MP-MoE compared with the corresponding OL-MoE baselines under the three training budgets. Across MMLU, BoolQ, HellaSwag, BBH, ARC-Easy, and ARC-Challenge, MP-MoE consistently improves downstream performance relative to standard routing at comparable training budgets. Under the 1B-token early pre-training, MP-MoE exhibits better sample efficiency, yielding higher average benchmark scores despite limited pre-training exposure. As the token budget increases to 5B, both methods improve, while MP-MoE maintains and often enlarges its advantage, suggesting that the proposed routing better leverages additional data. Notably, the smaller MP-MoE model, trained on 50B tokens, still outperforms the 1B-7B baseline, trained on 5B tokens, on multiple benchmarks, highlighting a favorable trade-off between model capacity and training duration enabled by improved routing.

To further demonstrate that MP-MoE outperforms other methods designed to enhance expert diversity, such as SimSMoE and HyperRouter (Do et al., 2023; 2025), we conduct additional experiments on the Flame-MoE (Kang et al., 2025). For a fair comparison, all models are configured to have 38M active parameters and are pre-trained under the same experimental setting across the three routing mechanisms. Both SimSMoE and HyperRouter are implemented following the default configurations reported in their original papers. Compared with the standard MoE baseline, MP-MoE achieves a 4.71% reduction in PPL, whereas SimSMoE achieves a 1.31% reduction and Hy-perRouter leads to a 20.62% increase in PPL. The resulting training loss and perplexity (PPL) curves are presented in Appendix B.3.

We acknowledge that the current version of our method is primarily designed for the pre-training stage. To provide an initial assessment of its effectiveness in supervised fine-tuning (SFT), we also conduct small-scale experiments, with detailed results reported in the Appendix B.4.

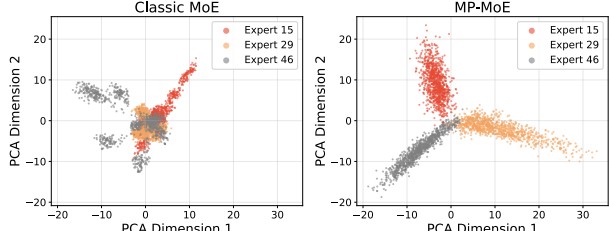

*Figure 3.* PCA plots for the expert output embeddings from the classic MoE (left) and MP-MoE (right). Compared with the classic MoE, MP-MoE exhibits substantially less overlap in the distributions of expert outputs. This suggests that MP-MoE more effectively promotes expert diversification during pre-training.

**Expert Diversification** To assess whether our method promotes expert diversification during pre-training, we analyze the separability of expert outputs at inference time. Specifically, we randomly select three experts from Layer 9 of an OLMoE model and sample 1,000 tokens randomly from the vocabulary. For each sampled token, we compute the corresponding output embedding produced by each of the three experts, and then apply PCA to these embeddings for visualization. The baseline and MP-MoE use the same layer index, expert identities, token set, and PCA procedure to ensure a controlled comparison. As shown in Figure 3, the baseline MoE exhibits substantial overlap among the projected expert outputs, indicating limited separation and higher redundancy. In contrast, MP-MoE yields more distinct clusters, suggesting that it encourages stronger expert specialization and reduces redundancy during pre-training.

Furthermore, to provide a quantitative description of expert diversity, we adopted the Centered Kernel Alignment (CKA) metric, as used in SimSMoE (Do et al., 2025), to quantitatively measure the similarity between experts. Specifically, we froze the parameters of the pre-trained MoE and MP-MoE models. For a sample of 10,000 tokens, we computed the output of each expert through both the MoE and MP-MoE layers. Based on these outputs, we calculated the expert-to-expert similarity using Linear-CKA. The table 3 below presents the average expert similarity across different layers.

*Table 3.* Expert-to-expert similarity measured by Linear-CKA. The reported values are given as the central estimate with the lower and upper 95th percentiles of expert similarity. Under identical training conditions, MP-MoE consistently yields lower inter-expert similarity than the classical MoE, indicating improved expert diversity.

| Method | Layer 2 | Layer 5 | Layer 9 |
|--------|---------|---------|---------|
| MoE | $0.43 \pm 0.04$ | $0.36 \pm 0.03$ | $0.37 \pm 0.07$ |
| **MP-MoE** | $\mathbf{0.31 \pm 0.03}$ | $\mathbf{0.28 \pm 0.02}$ | $\mathbf{0.30 \pm 0.05}$ |

**Ablation on MP-MoE Routing.** To isolate the contribution of the proposed routing mechanism, we conduct an ablation study on OLMoE-1B-7B to quantify the impact of the key design choices in MP-MoE. Replacing the Mahalanobis diversity objective with an $\ell_1$-norm criterion, which corresponds to replacing $\Sigma$ with the identity matrix $\mathbf{I}$, results in the largest performance drop ($-3.00\%$). This observation indicates that explicitly modeling inter-expert correlations is critical for selecting complementary experts. Substituting the covariance estimator $\Sigma$ with the raw co-occurrence matrix $\mathbf{C}$ also degrades performance ($-1.37\%$), suggesting that a properly normalized covariance estimate is necessary to capture meaningful expert dependencies. Finally, removing the warm-up schedule leads to a smaller yet consistent decrease ($-0.55\%$), implying that warm-up primarily improves optimization stability rather than constituting the main source of the observed gains. Detailed ablation results are provided in Appendix B.1.

Overall, MP-MoE delivers consistent performance gains across all evaluated benchmarks and training budgets. These results support four key conclusions: MP-MoE (i) improves average benchmark performance under matched compute budgets, (ii) exhibits stronger scaling behavior as the pre-training token budget increases, (iii) remains effective in smaller-capacity models, and (iv) preserves inference-time efficiency by employing the same deployment routing as the baseline.

## 5. Conclusion

In this paper, we have introduced Mahalanobis-Pruned Mixture-of-Experts (MP-MoE), a novel routing framework that approaches expert selection from the perspective of ensemble pruning. We have demonstrated that reformulating the routing process as a subset maximization problem governed by the Mahalanobis distance explicitly enhances expert diversity while preserving computational viability. Through a series of experiments, we have validated the effectiveness of our proposed method in mitigating representation collapse and its invariance to reasoning performance. In our experiments on OLMoE-1B-7B and OLMoE-0.2B-1B, MP-MoE consistently demonstrates superior performance compared to the baseline, achieving average accuracy im-

provements of up to 3.0 percentage points. Furthermore, it serves as a highly efficient, plug-and-play module that can be integrated into various sparse MoE architectures with negligible overhead. Promising future directions include: (i) extending the model to Mixture of Grouped Experts (Tang et al., 2025) to maximize inter-group diversity, and (ii) adapting the framework to MoE-LoRA (Li et al., 2024; Dou et al., 2024) architectures to enhance expert specialization during multi-task fine-tuning.

## Impact Statement

This work proposes MP-MoE, a diversity-aware routing framework for sparse Mixture-of-Experts models that improves pre-training quality under relatively matched compute budgets. A potential positive impact is achieving better training results, which may lower the barrier to research and deployment in resource-constrained settings and improve energy efficiency when comparable performance can be achieved with fewer training steps.

At the same time, improving the training efficiency and capability of language models can also amplify downstream risks associated with more capable generative systems, including misinformation generation, automation of harmful content at scale, and the reinforcement of societal biases present in training data. These risks depend strongly on deployment context and access controls.

We encourage practitioners to follow established safety and responsible deployment practices, including downstream evaluation for bias and misuse, usage policies and rate limits in deployment, and careful documentation of training data sources and licensing constraints when applicable. In particular, if copyrighted or sensitive data are used in training corpora, authors and users should acknowledge and address these considerations.

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

# A. Proof of Theorem 3.1

**Lemma A.1** (Approximation Guarantee for Greedy Selection). *Let $f : 2^U \to \mathbb{R}_{\geq 0}$ be a nonnegative, monotone set function, and let $OPT = \max_{|S| \leq k} f(S)$ denote the maximum value obtained by any set of size $k$. Let $S$ be the set selected by the Greedy algorithm. Then, the solution satisfies the following approximation guarantee:*

$$f(S) \geq \left(1 - e^{-\gamma_{U,k}}\right) \cdot OPT, \tag{6}$$

*where $\gamma_{U,k}$ is the submodularity ratio of $f$. Formally, $\gamma_{U,k}$ is defined as the minimum ratio of the marginal gain of a set to the marginal gain of its individual elements:*

$$\gamma_{U,k} = \min_{L \subseteq U, \, A:|A| \leq k, \, L \cap A = \emptyset} \frac{\sum_{x \in A} \left(f(L \cup \{x\}) - f(L)\right)}{f(L \cup A) - f(L)}. \tag{7}$$

While Lemma A.1 offers a general guarantee, the ratio $\gamma_{U,k}$ is typically intractable. We further provide a concrete bound for our Mahalanobis Ensemble Routing by considering the specific objective $f(S) = \boldsymbol{\mu}_S^\top \boldsymbol{\Sigma}_S^{-1} \boldsymbol{\mu}_S$. The following lemma bounds $\gamma_{U,k}$ via the restricted minimum eigenvalue of $\boldsymbol{\Sigma}$.

*Proof.* Proof can be found in Das & Kempe (2018). $\qquad\square$

**Lemma A.2** (Spectral Bound on Submodularity Ratio). *For $f(S) = \boldsymbol{\mu}_S^\top \boldsymbol{\Sigma}_S^{-1} \boldsymbol{\mu}_S$ and disjoint sets $L, A$, let $\boldsymbol{\Sigma}_{L \cup A}$ be partitioned naturally into blocks $\boldsymbol{\Sigma}_L, \boldsymbol{\Sigma}_A, \boldsymbol{\Sigma}_{LA}, \boldsymbol{\Sigma}_{AL}$. We define the residual statistics $\tilde{\boldsymbol{\mu}}$ and $\tilde{\boldsymbol{\Sigma}}$ (Schur complement) as:*

$$\tilde{\boldsymbol{\mu}} = \boldsymbol{\mu}_A - \boldsymbol{\Sigma}_{AL}\boldsymbol{\Sigma}_L^{-1}\boldsymbol{\mu}_L, \quad \tilde{\boldsymbol{\Sigma}} = \boldsymbol{\Sigma}_A - \boldsymbol{\Sigma}_{AL}\boldsymbol{\Sigma}_L^{-1}\boldsymbol{\Sigma}_{LA}.$$

*The submodularity ratio $\gamma_{U,k}$, which involves minimizing the term $\frac{\tilde{\boldsymbol{\mu}}^\top [\mathrm{diag}(\tilde{\boldsymbol{\Sigma}})]^{-1} \tilde{\boldsymbol{\mu}}}{\tilde{\boldsymbol{\mu}}^\top \tilde{\boldsymbol{\Sigma}}^{-1} \tilde{\boldsymbol{\mu}}}$, is bounded from below by the eigenvalues of $\boldsymbol{\Sigma}$:*

$$\gamma_{U,k} \geq \lambda_{\min}(\boldsymbol{\Sigma}, k + |U|) \geq \lambda_{\min}(\boldsymbol{\Sigma}), \tag{8}$$

*where $\lambda_{\min}(\boldsymbol{\Sigma}, k) \triangleq \min_{S:|S|=k} \lambda_{\min}(\boldsymbol{\Sigma}_S)$ denotes the smallest eigenvalue among all $k \times k$ principal submatrices of $\boldsymbol{\Sigma}$.*

*Proof.* **Part 1: Compute the submodularity ratio of $f(S)$.** Consider disjoint $L$ and $A$. Using the block inverse formula for $\boldsymbol{\Sigma}_{L \cup A}^{-1}$, one obtains the decomposition

$$f(L \cup A) = \boldsymbol{\mu}_L^\top \boldsymbol{\Sigma}_L^{-1} \boldsymbol{\mu}_L + \tilde{\boldsymbol{\mu}}^\top \tilde{\boldsymbol{\Sigma}}^{-1} \tilde{\boldsymbol{\mu}},$$

hence

$$f(L \cup A) - f(L) = \tilde{\boldsymbol{\mu}}^\top \tilde{\boldsymbol{\Sigma}}^{-1} \tilde{\boldsymbol{\mu}}.$$

For a singleton $a \in A$, the same calculation with $|A| = 1$ gives

$$f(L \cup \{a\}) - f(L) = \frac{\tilde{\mu}_a^2}{\tilde{\Sigma}_{aa}}.$$

Therefore, for any $L \subseteq U$ and $A$ with $|A| \leq k$ and $A \cap L = \emptyset$, the submodularity ratio of $f(S)$ becomes

$$\frac{\sum_{a \in A} \left(f(L \cup \{a\}) - f(L)\right)}{f(L \cup A) - f(L)} = \frac{\tilde{\boldsymbol{\mu}}^\top \left[\mathrm{diag}(\tilde{\boldsymbol{\Sigma}})\right]^{-1} \tilde{\boldsymbol{\mu}}}{\tilde{\boldsymbol{\mu}}^\top \tilde{\boldsymbol{\Sigma}}^{-1} \tilde{\boldsymbol{\mu}}}.$$

**Part 2: Lower bound by the minimum eigenvalue of a correlation matrix.** Let $\mathbf{D} = \mathrm{diag}(\tilde{\boldsymbol{\Sigma}})$ and define the correlation matrix $\tilde{\boldsymbol{\Sigma}}_\rho = \mathbf{D}^{-1/2} \tilde{\boldsymbol{\Sigma}} \mathbf{D}^{-1/2}$. Let $v = \mathbf{D}^{-1/2} \tilde{\boldsymbol{\mu}}$. Then

$$\tilde{\boldsymbol{\mu}}^\top \mathbf{D}^{-1} \tilde{\boldsymbol{\mu}} = \|v\|_2^2, \qquad \tilde{\boldsymbol{\mu}}^\top \tilde{\boldsymbol{\Sigma}}^{-1} \tilde{\boldsymbol{\mu}} = v^\top \tilde{\boldsymbol{\Sigma}}_\rho^{-1} v.$$

Hence the ratio equals $\frac{v^\top v}{v^\top \tilde{\boldsymbol{\Sigma}}_\rho^{-1} v}$. By Rayleigh–Ritz, $v^\top \tilde{\boldsymbol{\Sigma}}_\rho^{-1} v \leq \lambda_{\max}(\tilde{\boldsymbol{\Sigma}}_\rho^{-1})\, v^\top v = \frac{1}{\lambda_{\min}(\tilde{\boldsymbol{\Sigma}}_\rho)} v^\top v$, so

$$\frac{\tilde{\boldsymbol{\mu}}^\top \mathbf{D}^{-1} \tilde{\boldsymbol{\mu}}}{\tilde{\boldsymbol{\mu}}^\top \tilde{\boldsymbol{\Sigma}}^{-1} \tilde{\boldsymbol{\mu}}} \geq \lambda_{\min}(\tilde{\boldsymbol{\Sigma}}_\rho).$$

**Part 3: Relate $\lambda_{\min}(\tilde{\boldsymbol{\Sigma}}_\rho)$ to principal submatrices of $\boldsymbol{\Sigma}$.** We can eliminate the elements of $L$ one by one via residualization, each elimination replaces the current covariance by the covariance of residuals. By Lemma A.4, each such residualization step cannot decrease the smallest eigenvalue. After eliminating all of $L$, we obtain the Schur complement $\tilde{\boldsymbol{\Sigma}}$ corresponding to conditioning on $L$, and thus

$$\lambda_{\min}(\boldsymbol{\Sigma}_{L\cup A}) \leq \lambda_{\min}(\tilde{\boldsymbol{\Sigma}}).$$

Finally, normalizing $\tilde{\boldsymbol{\Sigma}}$ to unit variance gives $\tilde{\boldsymbol{\Sigma}}_\rho$, and Lemma A.3 yields

$$\lambda_{\min}(\tilde{\boldsymbol{\Sigma}}) \leq \lambda_{\min}(\tilde{\boldsymbol{\Sigma}}_\rho).$$

Combining the last two displays:

$$\lambda_{\min}(\tilde{\boldsymbol{\Sigma}}_\rho) \geq \lambda_{\min}(\boldsymbol{\Sigma}_{L\cup A}) \geq \lambda_{\min}(\boldsymbol{\Sigma}, |L \cup A|).$$

**Part 4: Combine the results.** By definition of $\gamma_{U,k}$ as the minimum over all $L \subseteq U$ and $A$ with $|A| \leq k$, we conclude

$$\gamma_{U,k} \geq \min_{L\subseteq U,\; A:\,|A|\leq k,\; A\cap L=\emptyset} \lambda_{\min}(\boldsymbol{\Sigma}, |L \cup A|) \geq \lambda_{\min}(\boldsymbol{\Sigma}, k + |U|).$$

The inequality $\lambda_{\min}(\boldsymbol{\Sigma}, k + |U|) \geq \lambda_{\min}(\boldsymbol{\Sigma})$ follows because any principal submatrix has the smallest eigenvalue at least that of the full matrix. $\qquad\square$

**Lemma A.3.** *Let $\boldsymbol{\Sigma}$ be the covariance matrix of $n$ zero-mean random variables $X_1, \ldots, X_n$, each with variance at most $1$. Let $\boldsymbol{\Sigma}_\rho$ be the corresponding correlation matrix (i.e., the covariance matrix after normalizing each $X_i$ to have unit variance). Then*

$$\lambda_{\min}(\boldsymbol{\Sigma}) \leq \lambda_{\min}(\boldsymbol{\Sigma}_\rho).$$

*Proof.* Proof can be found in Das & Kempe (2018). $\qquad\square$

**Lemma A.4.** *Let $\Sigma$ be the covariance matrix of $n$ random variables $X_1, \ldots, X_n$ with $\sigma_{ij} \triangleq \mathrm{Cov}(X_i, X_j)$ and $\sigma_{nn} = \mathrm{Var}(X_n) > 0$. For each $i \in \{1, \ldots, n-1\}$ define the residual*

$$\mathrm{Res}(X_i, X_n) \triangleq X_i - \beta_i X_n, \qquad \beta_i \triangleq \frac{\mathrm{Cov}(X_i, X_n)}{\mathrm{Var}(X_n)} = \frac{\sigma_{in}}{\sigma_{nn}}.$$

*Let $\Sigma'$ be the $(n-1) \times (n-1)$ covariance matrix of the residual variables $\mathrm{Res}(X_1, X_n), \ldots, \mathrm{Res}(X_{n-1}, X_n)$. Then*

$$\lambda_{\min}(\Sigma) \leq \lambda_{\min}(\Sigma').$$

*Proof.* For any $i, j \leq n-1$, by bilinearity of covariance,

$$\begin{aligned}
\Sigma'_{ij} &\triangleq \mathrm{Cov}(\mathrm{Res}(X_i, X_n), \mathrm{Res}(X_j, X_n)) \\
&= \mathrm{Cov}(X_i - \beta_i X_n,\; X_j - \beta_j X_n) \\
&= \mathrm{Cov}(X_i, X_j) - \beta_j \mathrm{Cov}(X_i, X_n) - \beta_i \mathrm{Cov}(X_n, X_j) + \beta_i\beta_j \mathrm{Var}(X_n) \\
&= \sigma_{ij} - \beta_j \sigma_{in} - \beta_i \sigma_{jn} + \beta_i\beta_j \sigma_{nn}.
\end{aligned}$$

Substituting $\beta_i = \sigma_{in}/\sigma_{nn}$ and $\beta_j = \sigma_{jn}/\sigma_{nn}$ gives

$$\Sigma'_{ij} = \sigma_{ij} - \frac{\sigma_{jn}}{\sigma_{nn}}\sigma_{in} - \frac{\sigma_{in}}{\sigma_{nn}}\sigma_{jn} + \frac{\sigma_{in}\sigma_{jn}}{\sigma_{nn}^2}\sigma_{nn} = \sigma_{ij} - \frac{\sigma_{in}\sigma_{jn}}{\sigma_{nn}}.$$

Equivalently, if we write $\Sigma$ in block form

$$\Sigma = \begin{pmatrix} \Sigma_{11} & s \\ s^\top & \sigma_{nn} \end{pmatrix}, \quad \Sigma_{11} \in \mathbb{R}^{(n-1)\times(n-1)}, \ s \in \mathbb{R}^{(n-1)\times 1}, \ s_i = \sigma_{in},$$

Then the above identity is exactly

$$\Sigma' = \Sigma_{11} - \frac{1}{\sigma_{nn}} ss^\top.$$

Define the rank-1 PSD matrix

$$D \triangleq \frac{1}{\sigma_{nn}} ss^\top \succeq 0,$$

so that the principal submatrix satisfies $\Sigma_{11} = \Sigma' + D$.

Let $\lambda_1' = \lambda_{\min}(\Sigma')$ and choose a (nonzero) eigenvector $e' \in \mathbb{R}^{n-1}$ such that

$$\Sigma' e' = \lambda_1' e'.$$

Now construct a vector $e \in \mathbb{R}^n$ by appending a carefully chosen last coordinate:

$$e \triangleq \begin{pmatrix} e' \\ e_n \end{pmatrix}, \qquad e_n \triangleq -\frac{1}{\sigma_{nn}} s^\top e' = -\frac{1}{\sigma_{nn}} \sum_{i=1}^{n-1} e_i' \sigma_{in}.$$

Then, using the block form of $\Sigma$,

$$\Sigma e = \begin{pmatrix} \Sigma_{11} & s \\ s^\top & \sigma_{nn} \end{pmatrix} \begin{pmatrix} e' \\ e_n \end{pmatrix} = \begin{pmatrix} \Sigma_{11} e' + s e_n \\ s^\top e' + \sigma_{nn} e_n \end{pmatrix}.$$

We first evaluate the last component:

$$s^\top e' + \sigma_{nn} e_n = s^\top e' + \sigma_{nn} \left( -\frac{1}{\sigma_{nn}} s^\top e' \right) = 0.$$

Thus, the last coordinate indeed cancels.

Next, for the first $(n-1)$ coordinates we substitute $e_n = -(1/\sigma_{nn}) s^\top e'$:

$$\begin{aligned} \Sigma_{11} e' + s e_n &= \Sigma_{11} e' - s \frac{1}{\sigma_{nn}} s^\top e' \\ &= \left( \Sigma_{11} - \frac{1}{\sigma_{nn}} ss^\top \right) e' \\ &= \Sigma' e'. \end{aligned}$$

Therefore,

$$\Sigma e = \begin{pmatrix} \Sigma' e' \\ 0 \end{pmatrix} = \begin{pmatrix} \lambda_1' e' \\ 0 \end{pmatrix}.$$

For any nonzero vector $x$, the Rayleigh quotient satisfies

$$\lambda_{\min}(\Sigma) = \min_{x \neq 0} \frac{x^\top \Sigma x}{x^\top x} \leq \frac{e^\top \Sigma e}{e^\top e}.$$

Using the expression for $\Sigma e$ obtained above,

$$e^\top \Sigma e = \begin{pmatrix} (e')^\top & e_n \end{pmatrix} \begin{pmatrix} \lambda_1' e' \\ 0 \end{pmatrix} = \lambda_1' (e')^\top e' = \lambda_1' \|e'\|_2^2.$$

Moreover,

$$e^\top e = \|e\|_2^2 = \|e'\|_2^2 + e_n^2 \geq \|e'\|_2^2.$$

Hence,

$$\frac{e^\top \Sigma e}{e^\top e} = \frac{\lambda_1' \|e'\|_2^2}{\|e\|_2^2} \leq \lambda_1'.$$

Combining the inequalities yields

$$\lambda_{\min}(\Sigma) \leq \frac{e^\top \Sigma e}{e^\top e} \leq \lambda_1' = \lambda_{\min}(\Sigma'),$$

which proves the claim. $\qquad\square$

**Lemma A.5.** *Assume variables are standardized so that $\Sigma$ is a correlation matrix, i.e., $\Sigma_{jj} = 1$ for all $j$. Let $\lambda \triangleq \lambda_{\min}(\Sigma, k)$. For any $k_0 = \Theta(k)$ such that $\frac{1}{\lambda} < k_0 < k$, and for sufficiently large $k$,*

$$f(S_{k_0}^\star) \geq f(S_k^\star) \cdot \Theta\left(\left(\frac{k_0}{k}\right)^{1/\lambda}\right).$$

*In particular, taking $k_0 = k/2$ gives*

$$OPT_{k/2} = f(S_{k/2}^\star) \geq OPT_k \cdot \Theta\left(\left(\tfrac{1}{2}\right)^{1/\lambda}\right).$$

*Proof.* Let $S = S_k^\star$ with $|S| = k$ and define

$$\alpha \triangleq \Sigma_S^{-1} \boldsymbol{\mu}_S \in \mathbb{R}^k.$$

Then

$$f(S) = \boldsymbol{\mu}_S^\top \Sigma_S^{-1} \boldsymbol{\mu}_S = \boldsymbol{\mu}_S^\top \alpha = \alpha^\top \Sigma_S \alpha.$$

Pick any $j \in S$, and let $R = S \setminus \{j\}$. Write the block partition

$$\Sigma_S = \begin{pmatrix} \Sigma_R & \Sigma_{Rj} \\ \Sigma_{jR} & \Sigma_{jj} \end{pmatrix}, \qquad \boldsymbol{\mu}_S = \begin{pmatrix} \boldsymbol{\mu}_R \\ p_j \end{pmatrix}.$$

By the Schur-complement, the marginal contribution of $j$ to $f$ satisfies

$$f(S) - f(R) = \frac{\left(p_j - \Sigma_{jR}\Sigma_R^{-1}\boldsymbol{\mu}_R\right)^2}{\Sigma_{jj} - \Sigma_{jR}\Sigma_R^{-1}\Sigma_{Rj}}. \tag{9}$$

Moreover, the $j$-th coordinate of $\alpha = \Sigma_S^{-1}\boldsymbol{\mu}_S$ is

$$\alpha_j = \frac{p_j - \Sigma_{jR}\Sigma_R^{-1}\boldsymbol{\mu}_R}{\Sigma_{jj} - \Sigma_{jR}\Sigma_R^{-1}\Sigma_{Rj}}, \tag{10}$$

hence combining (9)–(10) yields

$$f(S) - f(R) = \alpha_j^2 \cdot \left(\Sigma_{jj} - \Sigma_{jR}\Sigma_R^{-1}\Sigma_{Rj}\right). \tag{11}$$

The scalar in parentheses is the conditional variance $\mathrm{Var}(\mathrm{Res}(X_j, X_R))$. Since $\Sigma_{jj} = 1$ and conditioning cannot increase variance, we have

$$0 \leq \Sigma_{jj} - \Sigma_{jR}\Sigma_R^{-1}\Sigma_{Rj} \leq \Sigma_{jj} = 1.$$

Therefore (11) implies the key bound

$$f(S) - f(R) \leq \alpha_j^2. \tag{12}$$

Now choose $j$ minimizing $\alpha_j^2$, so $\alpha_j^2 \leq \|\alpha\|_2^2/k$. By the Rayleigh quotient,

$$f(S) = \alpha^\top \Sigma_S \alpha \geq \lambda_{\min}(\Sigma_S)\|\alpha\|_2^2 \geq \lambda \|\alpha\|_2^2,$$

because $\lambda = \lambda_{\min}(\Sigma, k) \leq \lambda_{\min}(\Sigma_S)$. Thus $\|\alpha\|_2^2 \leq f(S)/\lambda$, and hence

$$\alpha_j^2 \leq \frac{\|\alpha\|_2^2}{k} \leq \frac{f(S)}{k\lambda}.$$

Plugging into (12) gives

$$f(R) = f(S) - (f(S) - f(R)) \geq f(S) - \frac{f(S)}{k\lambda} = \left(1 - \frac{1}{k\lambda}\right) f(S).$$

Since $S^\star_{k-1}$ is optimal among sets of size at most $k - 1$,

$$f(S^\star_{k-1}) \geq f(R) \geq \left(1 - \frac{1}{k\lambda}\right) f(S^\star_k). \tag{13}$$

Then, we apply (13) repeatedly for sizes $i = k, k - 1, \ldots, k_0 + 1$ yields

$$f(S^\star_{k_0}) \geq f(S^\star_k) \cdot \prod_{i=k_0+1}^{k} \left(1 - \frac{1}{i\lambda}\right). \tag{14}$$

Moreover, let $t = \lceil 1/\lambda \rceil$. For $i > t$, we have $\frac{1}{i\lambda} \leq \frac{t}{i}$ and hence

$$1 - \frac{1}{i\lambda} \geq 1 - \frac{t}{i} = \frac{i-t}{i}.$$

Therefore the product in (14) satisfies

$$\prod_{i=k_0+1}^{k} \left(1 - \frac{1}{i\lambda}\right) \geq \prod_{i=k_0+1}^{k} \frac{i-t}{i}.$$

Most terms telescope, giving

$$\prod_{i=k_0+1}^{k} \frac{i-t}{i} = \prod_{r=1}^{t} \frac{k_0 - t + r}{k - t + r}.$$

For fixed $t$ and large $k$ with $k_0 = \Theta(k)$, the last product converges to $\left(\frac{k_0}{k}\right)^t$; hence

$$\prod_{i=k_0+1}^{k} \left(1 - \frac{1}{i\lambda}\right) \geq \Theta\left(\left(\frac{k_0}{k}\right)^t\right) \geq \Theta\left(\left(\frac{k_0}{k}\right)^{1/\lambda}\right).$$

Substituting back into (14) proves the lemma.

$\square$

**Proof of Theorem 3.1.** Let the ground set of experts be $\mathcal{E}$ (with $|\mathcal{E}| = m$). Denote by $\mathcal{E}_t$ the greedy set after $t$ iterations, so $\mathcal{E}_0 = \varnothing$ and $\mathcal{E}_k$ is the output. Let

$$OPT \triangleq \max_{S \subseteq \mathcal{E}, |S| \leq k} f(S), \qquad S^\star \in \arg \max_{|S| \leq k} f(S)$$

be an optimal size-$k$ solution.

**Part 1: Apply the greedy guarantee in terms of the submodularity ratio.** Since $f$ is assumed nonnegative and monotone, Lemma A.1 applies and yields

$$f(\mathcal{E}_k) \geq \left(1 - e^{-\gamma_{\mathcal{E},k}}\right) \cdot OPT, \tag{15}$$

where $\gamma_{\mathcal{E},k}$ is the submodularity ratio of $f$ on the ground set $\mathcal{E}$.

**Part 2: Lower bound $\gamma_{\mathcal{E},k}$ by a restricted minimum eigenvalue.** Now specialize to

$$f(S) = \boldsymbol{\mu}_S^\top \Sigma_S^{-1} \boldsymbol{\mu}_S,$$

where $\mu$ is the score vector and $\Sigma$ is the expert covariance matrix. Lemma A.2 shows that for any disjoint sets $L, A$ ( $|A| \leq k$ ), the ratio inside the definition of $\gamma_{\mathcal{E},k}$ satisfies

$$\frac{\sum_{x \in A}\big(f(L \cup \{x\}) - f(L)\big)}{f(L \cup A) - f(L)} \geq \lambda_{\min}\big(\Sigma, |L| + |A|\big).$$

In Lemma A.1, the set $L$ is instantiated as a greedy prefix $\mathcal{E}_t$ for some $t \in \{0, 1, \ldots, k-1\}$, hence $|L| = |\mathcal{E}_t| = t \leq k-1$, and the comparison set $A$ can be taken as a subset of $S^\star$ with $|A| \leq k$. Therefore, along the greedy trajectory, we always have

$$|L| + |A| \leq (k-1) + k \leq 2k,$$

and consequently

$$\gamma_{\mathcal{E},k} \geq \min_{t \in \{0,\ldots,k-1\}} \min_{\substack{A \subseteq \mathcal{E} \setminus \mathcal{E}_t \\ |A| \leq k}} \lambda_{\min}\big(\Sigma, |\mathcal{E}_t| + |A|\big) \geq \lambda_{\min}(\Sigma, 2k).$$

Plugging this into (15) gives the second line claimed in the theorem:

$$f(\mathcal{E}_k) \geq \big(1 - e^{-\lambda_{\min}(\Sigma, 2k)}\big) \cdot OPT. \tag{16}$$

**Part 3: Telescope.** Apply the second-line guarantee after $r = k/2$ greedy steps:

$$f(\mathcal{E}_{k/2}) \geq \big(1 - e^{-\lambda_{\min}(\Sigma, 2 \cdot (k/2))}\big) \cdot OPT_{k/2} = \big(1 - e^{-\lambda_{\min}(\Sigma, k)}\big) \cdot OPT_{k/2}.$$

By monotonicity of $f$, $f(\mathcal{E}_k) \geq f(\mathcal{E}_{k/2})$. Finally, by Lemma A.5 with $k_0 = k/2$ we have $OPT_{k/2} \geq OPT_k \cdot \Theta((1/2)^{1/\lambda_{\min}(\Sigma, k)})$. Combining the last three displays yields

$$f(\mathcal{E}_k) \geq \big(1 - e^{-\lambda_{\min}(\Sigma, k)}\big) \cdot \Theta\Big(\big(\tfrac{1}{2}\big)^{1/\lambda_{\min}(\Sigma, k)}\Big) \cdot OPT_k,$$

which is exactly the third line claimed.

**Part 4: Lower bound $\lambda_{\min}(\Sigma, k)$ by co-occurrence probabilities via Gershgorin.** Assume $\Sigma$ is the covariance matrix of the expert-selection indicator vector $Z = (Z_1, \ldots, Z_m)$, where $Z_j \in \{0, 1\}$ indicates whether expert $j$ is selected. Let

$$P_j \triangleq \mathbb{P}(Z_j = 1), \qquad P_{ij} \triangleq \mathbb{P}(Z_i = 1, Z_j = 1).$$

Then the entries of $\Sigma$ are

$$\Sigma_{jj} = \mathrm{Var}(Z_j) = P_j(1 - P_j), \qquad \Sigma_{ij} = \mathrm{Cov}(Z_i, Z_j) = P_{ij} - P_i P_j \quad (i \neq j).$$

Fix any set $S \subseteq \mathcal{E}$ with $|S| = k$ and consider the principal submatrix $\Sigma_S$. By Gershgorin's circle theorem, every eigenvalue $\lambda$ of $\Sigma_S$ lies in at least one disc

$$\Big\{z \in \mathbb{C} : |z - \Sigma_{jj}| \leq \sum_{i \in S, \, i \neq j} |\Sigma_{ij}|\Big\} \quad \text{for some } j \in S.$$

In particular, the smallest eigenvalue satisfies

$$\lambda_{\min}(\Sigma_S) \geq \min_{j \in S}\Big(\Sigma_{jj} - \sum_{i \in S, \, i \neq j} |\Sigma_{ij}|\Big) = \min_{j \in S}\Big(P_j(1 - P_j) - \sum_{i \in S, \, i \neq j} |P_{ij} - P_i P_j|\Big).$$

Since $\lambda_{\min}(\Sigma, k) = \min_{|S|=k} \lambda_{\min}(\Sigma_S)$, the above inequality implies that for any specific $k$-set $S$ (in particular, $S = \mathcal{E}_k$),

$$\lambda_{\min}(\Sigma, k) \geq \min_{j \in \mathcal{E}_k}\Big(P_j(1 - P_j) - \sum_{i \in \mathcal{E}_k, \, i \neq j} |P_{ij} - P_i P_j|\Big),$$

which is exactly the co-occurrence lower bound stated in the theorem.

**Part 5: Combine the results.** Combining (16) with the Gershgorin lower bound completes the proof of the theorem. $\qquad\square$

# B. Additional Experiments

## B.1. Ablation Study on MP-MoE Routing Design

All ablation experiments are conducted on the OLMoE-1B-7B model pre-trained for 1B tokens. The **Diversity Objective** column specifies whether routing is guided by a Mahalanobis-distance objective or an $\ell_1$-distance objective, which corresponds to using $\Sigma$ or identity matrix $\mathbf{I}$ in the routing process. The **Covariance Estimator** column indicates whether the expert co-occurrence matrix $\mathbf{C}$ is used directly as the covariance matrix, or whether the covariance is computed using the estimator in Eq. 5. The **Warm-up Strategy** column denotes whether training uses classic MoE routing during the first 1% of updates as a warm-up phase before switching to MP-MoE routing.

*Table 4.* Ablation study on MP-MoE routing design on OLMoE-1B-7B

| Variant | Diversity Objective | Covariance Estimator | Warm-up Strategy | Average Accuracy Gain (%) |
|---|---|---|---|---|
| Full MP-MoE | Mahalanobis | Covariance $\Sigma$ | Warm-up | 0 |
| w/o Diversity | $\ell_1$-norm | Covariance $\Sigma$ | Warm-up | -3.00 |
| w/o $\Sigma$ | Mahalanobis | Co-occurrence $\mathbf{C}$ | Warm-up | -1.37 |
| w/o Warm-up | Mahalanobis | Covariance $\Sigma$ | no Warm-up | -0.55 |

## B.2. Comparison of Expert Similarity Between Classic MoE and MP-MoE

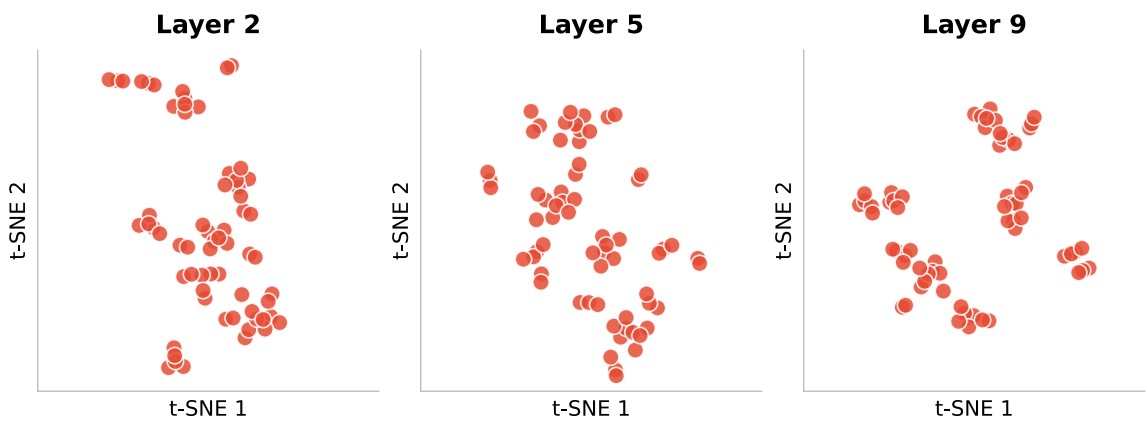

*(a)* Expert Output Similarity of MP-MoE

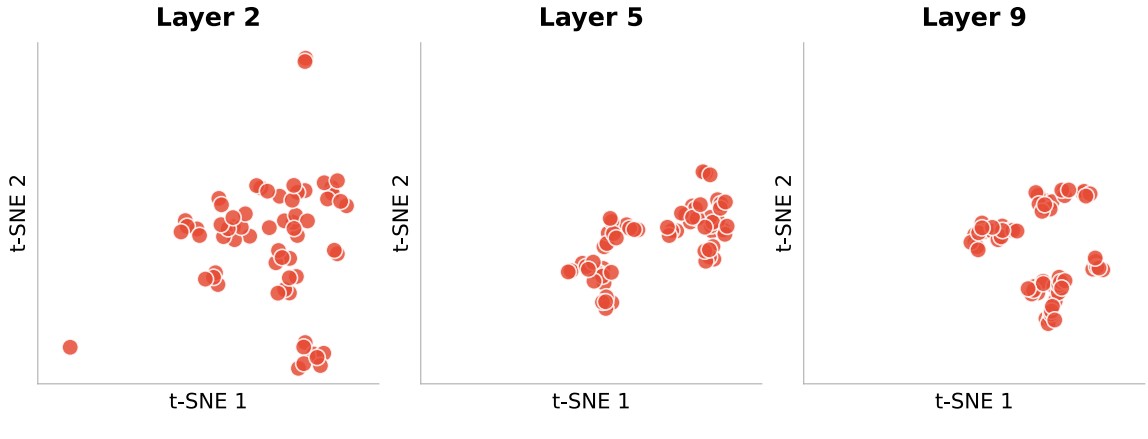

*(b)* Expert Output Similarity of MoE

*Figure 4.* Comparison of expert representational diversity. (a) MP-MoE exhibits greater expert diversity with distinct clusters, whereas (b) Classic MoE shows significant overlapping and redundancy under identical settings.

In this supplementary experiment, we further evaluate MP-MoE's ability to diversify expert outputs relative to the classic MoE. Following the protocol in Figure 2a, we randomly sample 1,000 tokens and compute, for both MP-MoE and the baseline, the cosine-similarity matrices of expert outputs at Layers 2, 5, and 9. We then flatten each similarity matrix to form an expert similarity vector for every expert. Using the same t-SNE configuration, we project these vectors into two dimensions for visualization. As shown in the figure, MP-MoE produces expert representations that are consistently more dispersed across all examined layers, indicating greater diversity among experts.

### B.3. Pre-Training Performance of Standard MoE, SimSMoE, HyperRouter, and MP-MoE

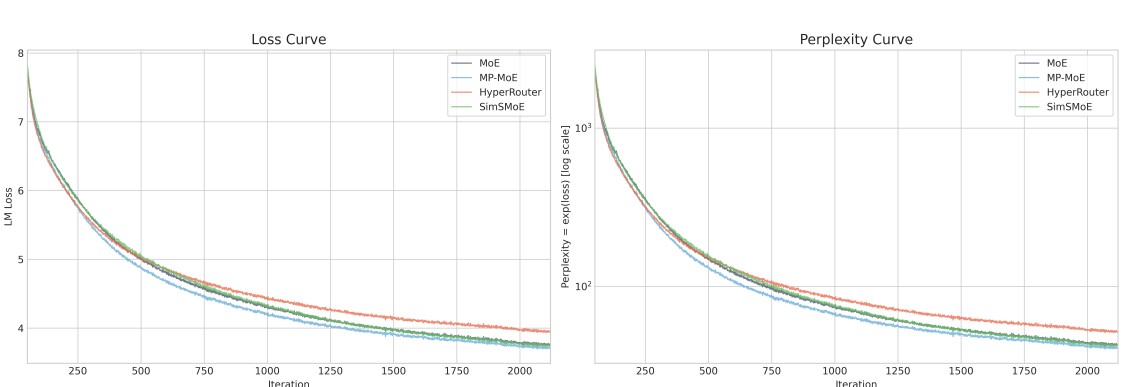

*Figure 5.* Training loss and perplexity (PPL) curves during pre-training. Compared with the standard MoE baseline, MP-MoE achieves a 4.71% reduction in PPL, whereas SimSMoE achieves a 1.31% reduction and HyperRouter leads to a 20.62% increase in PPL. These results further demonstrate the effectiveness and generalizability of MP-MoE.

### B.4. Supervised Fine-Tuning Performance of Standard MoE, SimSMoE, and MP-MoE

To provide a preliminary evaluation of MP-MoE in the supervised fine-tuning (SFT) stage, we conduct a small-scale experiment based on the Flame-MoE architecture. Specifically, we first pre-train a 98M/349M model and then fine-tune it on the Alpaca dataset. We compare three routing methods during fine-tuning, namely standard MoE, SimSMoE, and MP-MoE. The corresponding fine-tuning loss and perplexity (PPL) curves are shown in Figure 6.

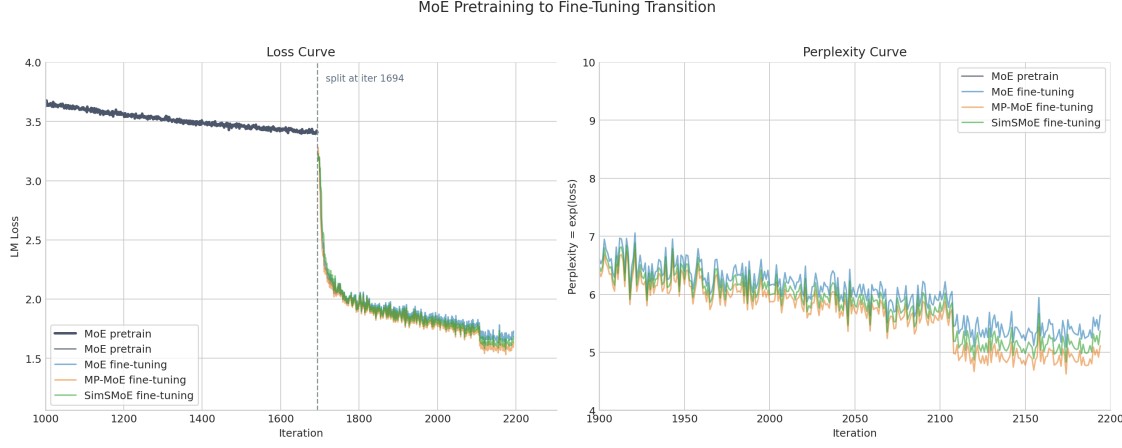

*Figure 6.* Training loss and perplexity (PPL) curves during fine-tuning. Compared with the standard MoE baseline, MP-MoE achieves a 9.31% reduction in PPL, while SimSMoE achieves a 4.90% reduction. These results further demonstrate the effectiveness and generalizability of MP-MoE in the SFT setting.

We further evaluate the fine-tuned models on several benchmark datasets, including COPA for commonsense causal reasoning, RTE for natural language inference, OpenBookQA for science question answering, WSC for pronoun resolution,

and HellaSwag for contextual commonsense completion. The results are reported in Table 5. Overall, MP-MoE consistently achieves the best performance across all evaluated benchmarks, further supporting its superiority over standard MoE and SimSMoE in the fine-tuning stage.

*Table 5.* Benchmark performance after supervised fine-tuning.

| Benchmark | Pretrain | MoE Fine-tuning | SimSMoE Fine-tuning | MP-MoE Fine-tuning |
|---|---|---|---|---|
| COPA | 0.520 | 0.570 | 0.570 | **0.620** |
| HellaSwag | 0.270 | 0.271 | 0.272 | **0.284** |
| OpenBookQA | 0.260 | 0.266 | 0.266 | **0.280** |
| RTE | 0.525 | 0.526 | 0.505 | **0.549** |
| WSC | 0.355 | 0.356 | 0.346 | **0.529** |

