# OpenReview forum: "Breaking the Echo Chamber: A Dynamic Ensemble Pruning Perspective on MoE"
_ICML.cc/2026/Conference — ICML 2026 regular_

### Official Review · Reviewer_CWhF · 2026-02-20

**Soundness:** 3
**Presentation:** 3
**Significance:** 3
**Originality:** 3
**Overall Recommendation:** 5
**Confidence:** 1

**Summary:**

This paper introduces a method based on Mahalanobis distance to solve the problem of the Echo Chamber in classic MoE models. The paper demonstrates the relationship between expert co-occurrence and expert output similarity. The paper is generally well-written and tackles a relevant problem, and the idea is intuitive and easy to follow.

**Compliance With Llm Reviewing Policy:**

Affirmed.

**Final Justification:**

Thank you for the detailed rebuttal. The authors provide CKA analysis for expert diversity, stronger baseline comparisons, and clearer justification of the method’s efficiency and originality. As my initial score already reflected a strong evaluation, I will not improve the overall rating. However, due to the endeavor of providing a clearer statement of the originality, I increase the sub-score of it.

**Key Questions For Authors:**

- The evaluation of expert diversity is a visualization. However, as solving the echo chamber is the main goal of this paper, why not use a proxy metric related to the expert diversity?

- There should be other MoE baselines to be compared.

**Limitations:**

Yes

**Strengths And Weaknesses:**

- Strengths:
1. *Soundness:* The paper is technically sound and methodologically reasonable. The core idea is intuitive and carefully validated through both empirical studies and reasonable analyses. The proposed module is easy to integrate and introduce acceptable overhead.

2. *Presentation:* The paper is well organized and clearly written. The motivation is straightforward, and the methodology is easy to follow.  The figures and tables are informative and help readers quickly grasp the improvements.

3. *Significance:* The echo chamber problem is a practical and important issue in modern MoE systems. As a plug-and-play solution, this work offers clear practical value for researchers and practitioners using MoE models. This work achieves good accuracy results with high expert diversity and acceptable computational overhead.

4. *Originality:* While the module itself is structurally simple, the work shows solid originality in identifying a clean and effective way to alleviate echo chamber effects.

- Weeknesses:

See Key Questions.

---

> ### Author Rebuttal · Authors · 2026-03-31
>
> We sincerely thank the reviewer for the strong endorsement, as well as for highlighting the importance of expert diversity and recognizing our work's contribution in this area. Detailed responses to your comments are provided below, and the manuscript will be revised accordingly in the camera-ready version.
>
> **Q1: The evaluation of expert diversity is a visualization.**
>
> We acknowledge that while the PCA visualizations of expert output embeddings in Figure 3  demonstrate that our method increases the representational differences between experts, they are not quantitative metrics. To address this, we have adopted the **Centered Kernel Alignment (CKA)** metric, as used in SimSMoE [2], to quantitatively measure the similarity between experts.
> Specifically, we froze the parameters of the pre-trained MoE and MP-MoE models. For a sample of 10,000 tokens, we computed the output of each expert through both the MoE and MP-MoE layers. Based on these outputs, we calculated the expert-to-expert similarity using Linear-CKA. The table below presents the average expert similarity across different layers.
>
> |Method|Layer 2|Layer 5|Layer 9|
> |:-|:-|:-|:-|
> |MoE|0.43±0.04|0.36±0.03|0.37±0.07|
> |**MP-MoE**|**0.31±0.03**|**0.28±0.02**|**0.30±0.05**|
>
>
> As shown in the table, the expert similarity decreases significantly under the MP-MoE router mechanism. This provides strong quantitative evidence that our Mahalanobis Ensemble Routing effectively enhances expert diversity.
>
> **Q2: Why not use a proxy metric related to the expert diversity?**
>
> We agree that certain metrics can measure expert similarity. As mentioned in Table 1 and the Introduction, methods like SimSMoE use CKA as a penalty term during training to minimize similarity between experts. However, explicitly calculating a "proxy metric" for expert similarity typically requires obtaining the latent embeddings of all experts.
> Since the MoE architecture is inherently designed to be sparse, computing the embeddings for *all* experts at every routing step would completely negate the computational efficiency of the MoE paradigm. Instead, by mathematically revealing the underlying relationship between the co-occurrence matrix and the similarity matrix, our method elegantly bypasses this expensive computation, allowing us to directly obtain a proxy for expert similarity without extracting full representations. We consider this efficient formulation to be one of our core contributions.
>
> **Q3: More MoE baselines to be compared.**
>
> To further demonstrate the potential and generalizability of MP-MoE, we evaluated it on the Flame-MoE [1] architecture and expanded our comparisons against strong MoE baselines, namely SimSMoE and HyperRouter [3]. For a fair comparison, we configured all models with 38M active parameters and conducted pre-training across these three routing mechanisms. Both SimSMoE and HyperRouter were implemented using the default settings from their original papers. The training loss and PPL curves are available here: https://anonymous.4open.science/r/MP-MoE-3F3F/figure/pretrain_loss_ppl.png , and the final training metrics are summarized in the table below：
>
> |Method|Train-End Loss|Train-End PPL|PPL Improvement vs MoE|
> |-|-:|-:|-:|
> |MoE|3.767|43.262|0.000%|
> |HyperRouter|3.954|52.162|-20.621%|
> |SimSMoE|3.754|42.694|1.313%|
> |**MP-MoE**|**3.719**|**41.225**|**4.708%**|
>
> Both the curves and the table show that our method consistently achieves superior performance compared to the baselines, demonstrating its overall effectiveness.
>
> **Q4: Regarding the originality of our paper:**
>
> We respectfully emphasize that a key contribution of our work is framing the MoE routing problem through the lens of ensemble learning. Additionally, to the best of our knowledge, we are the first to demonstrate the practical connection between the **co-occurrence matrix and expert similarity**. This insight offers an effective way to bypass the computationally expensive process of explicit representation extraction.
> Although our final optimization form is mathematically simple, we believe this does not diminish the paper's originality. Rather, it is precisely this simplicity and elegance that allows MP-MoE to serve as a **highly efficient, plug-and-play module** that can be easily adapted to various existing architectures.
>
> In summary, we hope the above responses successfully address your concerns and strengthen your confidence in supporting our work during the discussion and decision phases. Thank you for your time and constructive feedback.
>
> **References**
>
> [1] FLAME-MoE: A Transparent End-to-End Research Platform for Mixture-of-Experts Language Models. 2025
>
> [2] Simsmoe: Toward efficient training mixture of experts via solving representational collapse. 2025
>
> [3] Hyperrouter: Towards efficient training and inference of sparse mixture of experts. 2023

---

> > ### Author Rebuttal · Reviewer_CWhF · 2026-03-31
> >
> > Thank you for your reply and detailed clarification. I will take this into account and adjust the Originality.

---

> > > ### Author Response · Authors · 2026-04-01
> > >
> > > Thank you for the acknowledgment and thoughtful feedback. We are glad that our rebuttal resolved your concerns.

---

### Official Review · Reviewer_qQPC · 2026-03-08

**Soundness:** 3
**Presentation:** 2
**Significance:** 2
**Originality:** 2
**Overall Recommendation:** 2
**Confidence:** 3

**Summary:**

The paper introduces Mahalanobis-Pruned Mixture-of-Experts (MP-MoE) to address the issue of expert representation collapse. The authors claim that even when load-balancing  methods avoids expert idling, they can still process highly overlapping tokens and become redundant. To mitigate this, it replaces standard top-k routing with a subset selection mechanism based on the Mahalanobis distance. Specifically, it utilizes the expert co-occurrence matrix to estimate the covariance. Experiments on OLMoE show some gains over the baselines.

**Compliance With Llm Reviewing Policy:**

Affirmed.

**Final Justification:**

The rebuttal addressed most of my concerns and gave me further clarity about the submission. I have updated my soundness evaluation of the work accordingly. I will keep my overall recommendation score.

**Key Questions For Authors:**

Please read weakness.

**Strengths And Weaknesses:**

**Strength:**

1)	Solving MoE routing as ensemble pruning subset selection problem using Mahalanobis distance is indeed interesting.

2)	The use of expert co-occurrence as a proxy for expert correlation seems neat and efficient.


**Weakness:**

1)	The authors say “…At inference time, MP-MoE uses the same standard softmax top-k routing as the OLMoE baseline…”  (line 365). It creates mismatch between training-testing. While it helps achieve “no additional inference-time overhead”, it invalidates the paper’s method as a true routing mechanism and makes it simply a regularization trick during training.

2)	Fig 3 and 4 do show increased expert diversity but this validation is merely the result of applied penalty which is obvious. It doesn’t prove the experts' functional specialization.

3)	Theorem 3.1 borrows a mathematical proof from Das & Kempe (line 623). I believe it relies on the condition of stationarity of environment. However, as Algorithm 2 shows, environment is not static. The algorithm routes tokens to experts. Based on where tokens go, algorithm updates the co-occurrence matrix which updates token for next cycle. Could the reviewers clarify this misalignment?

4)	Table 1 criticizes several SOTA diversity-routing methods but they haven’t been compared in experiment design.

5)	What does Aux free means in Table 1? Later in Sec 3.3 line 313 “…motivating our practical choice to integrate traditional load balancing auxiliary losses.” Could the authors clarify this inconsistency?

---

> ### Author Rebuttal · Authors · 2026-03-31
>
> Thank you for your thoughtful feedback. Below, we provide point-by-point responses to your comments.
>
> **Q1: MP-MoE creates a mismatch between training and testing.**
>
> We would like to clarify that although the training-time selector in MP-MoE is not identical to the inference-time selector, this is an intentional **train-inference asymmetry** rather than a design flaw. During pre-training, MP-MoE employs a stronger, routing-aware mechanism explicitly to encourage functional specialization among experts, rather than a regularization technique. Because the experts have already cultivated diversity under this training mechanism, the model can seamlessly transition to the standard softmax top-k routing during inference, which maximizes computational efficiency, compatibility, and robustness. Furthermore, such train-inference asymmetry is a well-established practice in deep learning and has been adopted by influential MoE architectures like GShard [1] and Sparsely-Gated MoE [2]. Therefore, we respectfully suggest that this should be viewed as a standard design choice rather than a conceptual inconsistency.
>
> **Q2: Fig. 3 and 4 do show increased expert diversity but this validation is merely the result of applied penalty.**
>
> We first clarify that our method is **not** penalty-based: during pre-training, we formulate expert selection as an **optimal subset selection** problem rather than adding an extra regularization term as in Guo et al. (2025) or SimSMoE. In each forward pass, we use the expert co-occurrence matrix to estimate inter-expert similarity and select high-scoring yet diverse experts, which helps prevent experts from becoming overly similar due to repeatedly receiving overlapping tokens. Moreover, our training loss curves consistently outperform the baseline, and to further quantify expert specialization, we additionally report **CKA**, which is also the main metric used in SimSMoE.
>
> |Method|Layer 2|Layer 5|Layer 9|
> |:-|:-|:-|:-|
> |MoE|0.43±0.04|0.36±0.03|0.37±0.07|
> |**MP-MoE**|**0.31±0.03**|**0.28±0.02**|**0.30±0.05**|
>
> As shown above, after training with MP-MoE, expert similarity decreases substantially.
>
> **Q3: Whether Theorem 3.1 still holds when Algorithm 2 updates the routing statistics dynamically.**
>
> We agree that Theorem 3.1 relies on a stationarity assumption, but this does not affect our guarantee because the co-occurrence matrix is updated **batch-wise**. For each batch, it is fixed before routing and depends only on accumulated historical statistics, so Theorem 3.1 still ensures that the greedy algorithm is approximately optimal **within each batch**.
>
> **Q4: Compare with several SOTA diversity-routing methods.**
>
> Thank you for this comment. To further validate the effectiveness of our method, we have included comparisons against strong diversity-routing baselines such as SimSMoE[4] and HyperRouter[3]. Specifically, we conducted pre-training on a lightweight 38M Flame-MoE[5] architecture across these different routing mechanisms. The pre-training loss and perplexity (PPL) curves can be viewed at this anonymous link: `https://anonymous.4open.science/r/MP-MoE-3F3F/figure/pretrain_loss_ppl.png`. The final training metrics are summarized in the table below:
>
> |Method|Train-End Loss|Train-End PPL|PPL Improvement vs MoE|
> |-|-:|-:|-:|
> |MoE|3.76|43.26|0.00%|
> |HyperRouter|3.95|52.16|-20.62%|
> |SimSMoE|3.75|42.69|1.31%|
> |**MP-MoE**|**3.71**|**41.22**|**4.70%**|
>
> The empirical results clearly show that MP-MoE significantly outperforms the baselines in both the loss curves and the final PPL metrics.
>
> **Q5: Clarify the inconsistency between “Aux-free” in Table 1 and our use of auxiliary load-balancing losses in Section 3.3**
>
> We clarify that **Aux. Free** means our method introduces no additional diversity-specific regularization term into the training loss, such as similarity- or CKA-based penalties. Our focus is expert specialization, while load balancing addresses under-trained experts; these two goals are complementary. Since load balancing is a standard component in MoE and, as suggested by Theorem 3.1, can help the greedy algorithm better approximate the optimum, we keep it in both the baseline and all comparisons. In contrast, our method does not add any extra diversity loss.
>
> In summary, we hope the above responses address your concerns and help you re-evaluate our work. Feel free to contact us if you have any other questions.
>
> **References:**
>
> [1] GShard: Scaling Giant Models with Conditional Computation and Automatic Sharding. 2021
>
> [2] Outrageously Large Neural Networks: The Sparsely-Gated Mixture-of-Experts Layer. 2017
>
> [3] HyperRouter: Towards Efficient Training and Inference of Sparse Mixture of Experts. 2023
>
> [4] SimSMoE: Toward Efficient Training Mixture of Experts via Solving Representational Collapse. 2025
>
> [5] FLAME-MoE: A Transparent End-to-End Research Platform for Mixture-of-Experts Language Models. 2025

---

> > ### Author Rebuttal · Reviewer_qQPC · 2026-04-04
> >
> > We thank the reviewers for their detailed response. It addresses some of my concerns. I do still have a few remaining concerns:
> >
> > a) For W1, my initial concern was that the model has no mechanism to penalize any distribution shift. If the authors consider it a design choice, I request that they explicitly mention it in the discussion as a limitation or provide some solution on how the method can be adapted.
> >
> > b) For W2, I acknowledge the metric being used in a peer-reviewed work, but I believe CKA captures feature orthogonalization, which is one of the geometric proxies for specialization. I therefore request that authors lower their claim to "distinct specialization".
> >
> > c) For W3, this is my biggest concern. I believe the authors use a static maximization proof (Das & Kempe) to validate an online process. I request that the authors explicitly state the assumptions made and the limitations in the discussion.
> >
> > d) For W4, my concerns are fully addressed.
> >
> > e) My W5 concern is addressed. Please revise to include "diversity" as aux free in the revision.
> >
> > Overall, while mostly my concerns are resolved, my remaining concerns on claim and theorem scope persist.

---

> > > ### Author Response · Authors · 2026-04-04
> > >
> > > We thank the reviewer for the follow-up questions and constructive feedback. Below, we provide point-by-point responses and clarify your remaining concerns.
> > >
> > > Response to (a) / W1:
> > >
> > > Thank you for clarifying this point. We may have misunderstood your previous concerns.
> > >
> > > By **distribution shift**, we refer to the mismatch between the data distribution encountered during training and that encountered during inference or downstream deployment. This is indeed an important and widespread issue in large-model training. However, **handling distribution shift is not the primary objective of our work**. Our method is designed to improve **expert diversity** during MoE training, rather than to explicitly detect or penalize train–test distribution mismatch.
> > >
> > > That said, we agree it would be helpful to state this more explicitly in the discussion section as a limitation/design scope. In practice, distribution shift can be addressed by standard techniques such as **domain-specific fine-tuning**, **importance weighting**, and related adaptation methods. These approaches are fully compatible with **MP-MoE** and can be combined with our method without conflict.
> > >
> > > More broadly, we believe that training with an **expert similarity matrix** and using the resulting routing mechanism at inference can provide a relatively robust way to mitigate the effect of distribution variation, although this is not the main claim of the paper. We will clarify this point more carefully in the final version.
> > >
> > > Response to (b) / W2:
> > >
> > > We would like to clarify that we do not claim to propose a novel metric for "distinct specialization." For our definition of specialization, we strictly adhere to the standard established in prior work—namely, measuring the similarity of expert outputs, where CKA serves as one such similarity measure. The primary contribution of our work lies **not** in redefining specialization metrics, but rather in **improving expert specialization through the lens of ensemble learning**. The rationale for utilizing the co-occurrence matrix to reflect expert similarity is that calculating actual output similarity requires a complete forward pass. Our approach elegantly circumvents this computational step, allowing the co-occurrence matrix to serve **as a highly efficient proxy** for measuring expert similarity.
> > >
> > > Response to (c) / W3:
> > >
> > > We fully understand your concern regarding the use of a static maximization proof (Das & Kempe) to validate an online training process. We wish to clarify that we have never denied the presence of a static assumption, as the model's training process inherently **involves discrete, batch-wise updates**. Specifically, our algorithm relies on the co-occurrence matrix obtained from the **previous batch** to perform its current update. We apply this theorem to demonstrate that we are approximating the optimal solution locally at the **per-batch level**, rather than claiming a global optimum for the entire online process. Consequently, our formulation does not require extra dynamic assumptions; the static conditions are entirely sufficient for our batch-wise context. We hope this addresses your core concern. We will explicitly state these assumptions and emphasize the batch-wise nature of the theorem in the revised manuscript.
> > >
> > >
> > > Response to (e) / W5:
> > >
> > > We thank you for this suggestion and will make the corresponding modifications to the table.
> > >
> > > In summary, we will carefully incorporate all the modifications and clarifications discussed during this rebuttal into the camera-ready version.
> > >
> > > We hope our explanations have fully addressed your concerns, especially regarding the theoretical aspects, and we kindly ask you to re-evaluate our work in light of these clarifications.

---

### Official Review · Reviewer_PJno · 2026-03-09

**Soundness:** 3
**Presentation:** 3
**Significance:** 3
**Originality:** 3
**Overall Recommendation:** 4
**Confidence:** 4

**Summary:**

This paper proposes a training optimization method called MP-MoE that reduces expert overlap by optimizing the Mahalanobis distance between MoE experts. With a slight increase in training cost, it achieves improvements over the original training method across multiple tasks.

**Compliance With Llm Reviewing Policy:**

Affirmed.

**Final Justification:**

The authors' rebuttal effectively addressed my concerns, and I have raised my score to reflect this.

**Key Questions For Authors:**

1. Regarding Weakness 1, does MP-MoE similarly improve model performance on specific domains? For example, mathematical tasks (MATH, GSM8K), coding tasks (HumanEval, MBPP)? Besides relative to baselines, does it also show significant improvement relative to existing methods (Auxiliary-loss-free[1], SimSMoE[2])?
2. Regarding Weakness 2, could you provide an analysis of the training process, including Loss curves/PPL comparisons against the baseline and the evolution of Mahalanobis patterns, to more clearly demonstrate how MP-MoE operates?
3. Given that the experiments primarily focus on the pre-training phase, could you provide SFT results based on some open-source general MoE base models to better demonstrate the practical utility of MP-MoE?

While I appreciate the solid theoretical analysis provided by the authors, the experimental design is relatively simplistic, raising concerns about the method's generalizability to other models and application scenarios. Nevertheless, if the authors can strengthen the empirical evaluation, I believe this has the potential to be a valuable contribution.

[1] Auxiliary-Loss-Free Load Balancing Strategy for Mixture-of-Experts
[2] SimSMoE: Solving Representational Collapse via Similarity Measure

**Limitations:**

The authors are encouraged to discuss the limitations of their method, specifically regarding its applicability in SFT scenarios and the additional inference overhead introduced. Clarifying these constraints would help readers better understand the trade-offs and practical applicability of MP-MoE.

**Strengths And Weaknesses:**

**Strengths:**
1. The paper's motivation is well-founded. Through experiments, it proves that expert co-occurrence during MoE training leads to information redundancy, then optimizes based on this insight, providing inspiration for future work
2. Results on OLMoE show that MP-MoE can outperform the original baseline on multiple tasks under the same training settings, and visualization results also demonstrate that expert redundancy is significantly reduced in models trained with MP-MoE
3. Rigorous theoretical proofs demonstrate the reliability of this work

**Weaknesses:**
1. The experimental section is relatively brief. The authors only compared OLMoE's performance on general tasks relative to the original model, lacking results on other MoE architectures, comparisons with other existing optimization methods, and results on specific domains
2. Missing analysis of the training process. The authors should provide changes in model Loss curves/PPL relative to the original baseline during training, as well as changes in Mahalanobis patterns, to more clearly demonstrate how MP-MoE operates

---

> ### Author Rebuttal · Authors · 2026-03-31
>
> We sincerely thank the reviewer for acknowledging our motivation and theoretical contributions. We provide a point-by-point response to your concerns below.
>
> **Q1: Does MP-MoE similarly improve model performance on specific domains?**
>
> We acknowledge that our current computational budget has not yet allowed us to pre-train a truly massive foundation model, which likely limits the gains we currently observe in highly specialized domains. For example, on GSM8K, both the baseline and MP-MoE achieve near-zero scores. That said, MP-MoE already shows clear improvements over the baselines on HumanEval for both the 1B and 7B models, and we believe that scaling up model size and training data will further unlock its potential across broader domains.
>
> **Q2: Does it also show significant improvement relative to existing methods?**
>
> To further validate the effectiveness of our method, we have included comparisons against strong baselines such as **SimSMoE**[2] and **HyperRouter**[3]. Specifically, we conducted pre-training on a lightweight Flame-MoE[1] architecture across these different routing mechanisms, setting the active parameter count to 38M. The pre-training loss and perplexity (PPL) curves can be viewed at this anonymous link: https://anonymous.4open.science/r/MP-MoE-3F3F/figure/pretrain_loss_ppl.png. The final training metrics are summarized in the table below:
>
> |Method|Train-End Loss|Train-End PPL|PPL Improvement vs MoE|
> |-|-:|-:|-:|
> |MoE|3.76|43.26|0.00%|
> |HyperRouter|3.95|52.16|-20.62%|
> |SimSMoE|3.75|42.69|1.31%|
> |**MP-MoE**|**3.71**|**41.22**|**4.70%**|
>
> The empirical results clearly show that MP-MoE significantly outperforms the baselines in both the loss curves and the final PPL metrics. Regarding Auxiliary-loss-free [4], its focus on load-balancing is distinct from our goal of expert functional specialization. Therefore, rather than using it as an empirical baseline, we will include a discussion of it in the camera-ready Introduction.
>
> **Q3: Provide an analysis of the training process?**
>
> As requested, we have provided the pre-training loss and PPL curves (please refer to Q2). These curves demonstrate that our method maintains a consistent advantage over the baselines throughout the training process. Furthermore, to more clearly illustrate the impact of the Mahalanobis patterns, we evaluated the **CKA similarity** between experts after training concluded:
>
> |Method|Layer 2|Layer 5|Layer 9|
> |:-|:-|:-|:-|
> |MoE|0.43±0.04|0.36±0.03|0.37±0.07|
> |**MP-MoE**|**0.31±0.03**|**0.28±0.02**|**0.30±0.05**|
>
> As shown in the table, MP-MoE significantly reduces expert similarity compared to the standard MoE baseline. This quantitative reduction serves as strong evidence for the effectiveness of our Mahalanobis distance-based pruning approach in promoting expert diversity.
>
> **Q4: Provide SFT results based on some open-source general MoE base models**
>
> We acknowledge that our current method is primarily focused on the pre-training stage. To address this concern, we additionally conducted a small-scale experiment to evaluate the performance of MP-MoE during supervised fine-tuning (SFT). Specifically, we first pre-trained a 98M-349M model based on Flame-MoE, and then fine-tuned it on the Alpaca dataset. We compared the fine-tuning performance of three routing methods: standard MoE, SimSMoE, and MP-MoE. The fine-tuning loss and PPL curves are available at the following link:
> https://anonymous.4open.science/r/MP-MoE-3F3F/figure/fine_tune_loss_ppl.png. The final fine-tuning loss and PPL are reported below:
>
> |Method|Train-End Loss|Train-End PPL|PPL Improvement vs MoE|
> |-|-:|-:|-:|
> |MoE|1.73|5.63| 0.00%|
> |SimSMoE|1.68|5.36| 4.80%|
> |**MP-MoE**|**1.63**|**5.11**|**9.31%**|
>
> These results indicate that MP-MoE consistently outperforms the baselines in both the loss curve and PPL metrics during fine-tuning, further demonstrating the effectiveness of our method.
>
> **Q5: Regarding Limitations**
>
> We thank the reviewer for pointing this out. While our experiments show MP-MoE's potential in fine-tuning, we note that the small sample sizes in this phase can prevent the co-occurrence matrix from accurately reflecting expert similarities, potentially causing instability. We will add a discussion of this limitation to the camera-ready version.
>
> In summary, we hope our responses successfully address your concerns and provide strong grounds for you to reconsider your evaluation of our paper. Please feel free to let us know if you have any further questions during the discussion phase.
>
> **References**
>
> [1] FLAME-MoE: A Transparent End-to-End Research Platform for Mixture-of-Experts Language Models. 2025
>
> [2] Simsmoe: Toward efficient training mixture of experts via solving representational collapse. 2025
>
> [3] Hyperrouter: Towards efficient training and inference of sparse mixture of experts. 2023
>
> [4] Auxiliary-loss-free load balancing strategy for mixture-of-experts. 2024

---

> > ### Author Rebuttal · Reviewer_PJno · 2026-04-01
> >
> > Thanks for the authors’ response. My concerns have been addressed. I will reconsider my score, and I also look forward to seeing the effectiveness of this work in large-scale, industrial model training settings.

---

> > > ### Author Response · Authors · 2026-04-01
> > >
> > > We sincerely thank you for reconsidering our work and updating the score. We will continue to explore the potential of this work in large-scale scenarios.

---

### Official Review · Reviewer_S5JL · 2026-03-23

**Soundness:** 3
**Presentation:** 3
**Significance:** 4
**Originality:** 4
**Overall Recommendation:** 5
**Confidence:** 4

**Summary:**

The paper focuses on

**Compliance With Llm Reviewing Policy:**

Affirmed.

**Final Justification:**

The responses to the rebuttal acknowledgement addressed the reviewer's remaining concerns and the paper should be revised accordingly. Reviewer considered computational limitations with conducting LLM experiments and the authors provide results for two MoE backbones (OLMoE and Flame-MoE). Revising score upward.

**Key Questions For Authors:**

See above.

**Limitations:**

Authors do not explicitly discuss limitations. Paper does note the additional computational overhead with MP-MoE. Additional discussion is needed on whether benefits will be preserved after fine-tuning.

Authors discuss potential negative impact related to MoE.

**Strengths And Weaknesses:**

**Soundness**
- Claims are mostly well-supported (ensemble expert pruning, approximations of Mahalanobis-based measures during training and experimental results). Authors link top-K MoE expert selection to ensemble pruning, motivate/justify their use of Mahalanobis distance for pruning, leverage expert co-occurrence to construct inter-expert covariance matrix and use Cholesky factorisation to simplify estimation of expert covariance matrix recursively during training.
  - Reviewer did not go over details of Theorem 3.1 proof in Appendix.

- Experiments are well designed. Models are trained from scratch, so the same training conditions are maintained across MoE and MP-MoE. Information about computing resources is provided. Results are averaged over at least three runs. Diverse tasks are evaluated.
  - Include measures of variation/stats and indicate which test datasets this is applicable to.

- Paper includes ablation analysis (OLMoE-1B-7B model pre-trained for 1B tokens) with the substitution of the Mahalanobis distance with l1-norm and replacement of the covariance estimator with raw co-occurrence matrix. However, only aggregate results are provided and do not present the results by dataset as in Table 2.

- Inclusion of complexity analysis for the additional computational overhead with MP-MoE.

- Need additional information to better demonstrate how the experts develop distinct specialisation or quantify expert diversity in MoE vs. MP-MoE, not simply showing one example from MP-MoE with limited details.
  - *"...effectively trapping experts in an ‘echo chamber’ where they fail to develop distinct specializations."*
  - What does "the vocabulary" in "sample 1,000 tokens randomly from the vocabulary" mean? Which task is it?

**Presentation**
- Paper is generally well-written and illustrated. Overall narrative is generally easy to follow as authors define the problem, cite relevant literature and explain/provide justification for their technical choices/methods.
  - Additional details needed for clarity/better comprehension. Figure/table captions need to be more descriptive to understand content with minimal reliance on information in the main text. Some notations need to be defined, including in captions and algorithms. Acronyms need to be defined when first used in the main text (e.g., *“Building on this, we introduce MP-MoE…”*). See below.
- The paper positions the work within the context of prior literature and discusses how the proposed routing mechanism (during training only) differs with prior work.
  - Table 1: Include references of prior work in the table, not just algorithm names, to enhance readability. Some references are provided in the main text.

- Authors use an open source model OLMoE (assuming this is from Allen AI, include reference) and release code, which potentially could facilitate reproducing results. Reviewer only briefly looked at code and did not run. Information in the Algorithms table needs to be better described to reproduce methods if relying exclusively on this information.
  - Define all notations in algorithms table.
  - Main text: $p_j - \alpha_j^\top z$

- Figure 1: Caption needs to be more descriptive to walkthrough content.
   - To minimise confusion, state that the MP-MoE content in Figure 1 is applicable only to training and MP-MoE uses the same standard softmax top-k routing as OLMoE baseline during inference.
   - Figure 1 indicates the co-occurence matrix is "Stored in Memory" and "Update". What does "Update" mean?
   - Clarify how router probabilities (P), co-occurrence matrix (C), experts covariance matrix (\Sigma) and Mahalanobis score (P') in the figure are related. While this information is in various areas in the text, providing information in the figure caption will help understanding the figure content.

- Figure 2: More details about the input/output or task.
  - What is Expert Correspondence in Fig 2a? Need a similar figure for MoE for comparison.
  - Figure 2b requires better description to understand content.
- Figure 3: More details about the input/output or task.
  - Need other visual examples to strengthen results.

- Benchmarks and Metrics: Include description of each dataset task and related performance measure for audience that is not familiar with the datasets.
- Table 2
  - Include more details about the MoE architecture (total # of experts and # of selected experts).
  - Need measure of variation as results are averaged across 3 runs.

**Significance**
- The paper addresses limitations of the (top-K) routing mechanism in MoE in LLM applications and could influence future research in improving MoE pretraining (potentially beyond LLMs) to enhance expert diversity.
- The scope of the work is limited to pretraining only as no additional fine-tuning is performed for the downstream tasks. It is not clear if the general benefits will be preserved/applicable after fine-tuning.

**Originality**
- The work offers a novel combination of existing techniques and presents a new method to enhance the router training in MoE. The paper contributions are clearly distinguished from closely related literature.
- Paper needs to provide additional details about OLMoE to better highlight which aspect of the router mechanism is encapsulated in OLMoE (similar to Table 1) for better contrast.

---

> ### Author Rebuttal · Authors · 2026-03-31
>
> Thank you for recognizing our contribution to improving expert diversity. Below, we respond to your comments in the following sections.
>
> **Writing and presentation:**
>
> > What does "the vocabulary" in "sample 1,000 tokens randomly from the vocabulary" mean? Which task is it?
>
> We use held-out validation data from `fineweb-edu-tokenized/sample-10BT` [1], sample sequences and tokens from it, and compute the corresponding expert outputs. The goal is to test whether different experts produce distinct outputs for the same input, thereby reflecting expert diversity.
>
> > "Some notations need to be defined, including in captions and algorithms."; "Figure 1: Caption needs to be more descriptive to walkthrough content."; "Figure 2b requires better description to understand content."; "Figure 3: More details about the input/output or task."; "Some notations need to be defined, including in captions and algorithms."
>
> Thank you for these suggestions. In the camera-ready version, we will clarify the figures, notation, formulas, and experimental settings, including the components in Figure 1, the CKA explanation in Figure 2b, and the expert outputs in Figure 3. We will also expand the **Benchmarks and Metrics** section with brief descriptions of each benchmark.
>
> > "Figure 1 indicates the co-occurence matrix is "Stored in Memory" and "Update". What does "Update" mean?"
>
> Here, "Update" means that the matrix we use to characterize expert diversity is updated continuously during training. This allows it to adapt more effectively to the evolving diversity structure of experts throughout training. We will add a clearer explanation of this in the final version.
>
> > Clarify how router probabilities (P), co-occurrence matrix (C), experts covariance matrix (\Sigma) and Mahalanobis score (P') in the figure are related.
>
> The router probabilities (\(P\)) are the expert scores produced by the router. The co-occurrence matrix (\(C\)) records how often pairs of experts are activated together for tokens. The expert covariance matrix (\(\Sigma\)) is computed from the co-occurrence matrix and captures inter-expert covariance. The Mahalanobis score (\(P'\)) is the diversity-aware expert score obtained after correcting the router scores using the expert covariance matrix. We agree that summarizing these relationships in the figure caption would improve readability, and we will add this clarification in the camera-ready version.
>
> **Experiment**
>
> > Need additional information to better demonstrate how the experts develop distinct specialisation or quantify expert diversity in MoE vs. MP-MoE
>
> To address this concern, we additionally use **CKA** to measure similarity between experts, which is also a core quantitative metric used in SimSMoE [2].
>
> |Method|Layer 2|Layer 5|Layer 9|
> |:-|:-|:-|:-|
> |MoE|0.43±0.04|0.36±0.03|0.37±0.07|
> |**MP-MoE**|**0.31±0.03**|**0.28±0.02**|**0.30±0.05**|
>
> As shown above, after training with MP-MoE, expert similarity decreases substantially. This supports our claim that the subset-selection strategy encourages the learned experts to exhibit stronger **functional specialization**.
>
> > It is not clear if the general benefits will be preserved/applicable after fine-tuning.
>
> To address this concern, we conducted a small-scale experiment to evaluate MP-MoE during fine-tuning. Specifically, we first pre-trained a 98M-349M model based on Flame-MoE[3], and then fine-tuned it on the Alpaca dataset. We compared three routing methods: standard MoE, SimSMoE, and MP-MoE. The results show that MP-MoE achieves a clearly faster decrease in training loss than both SimSMoE and standard MoE, further demonstrating the effectiveness of our method. The fine-tuning loss and PPL curves are available at:
> https://anonymous.4open.science/r/MP-MoE-3F3F/figure/fine_tune_loss_ppl.png
>
> The final fine-tuning loss and PPL are reported below:
>
> |Method|Train-End Loss|Train-End PPL|PPL Improvement vs MoE|
> |-|-:|-:|-:|
> |MoE|1.73|5.63| 0.00%|
> |SimSMoE|1.68|5.36| 4.80%|
> |**MP-MoE**|**1.63**|**5.11**|**9.31%**|
>
> These results show that MP-MoE consistently outperforms the baselines in both the loss curve and PPL during fine-tuning.
>
> **Limitations**
>
> > Additional discussion is needed on whether benefits will be preserved after fine-tuning.
>
> Thank you for this suggestion. We will add a discussion in the camera-ready version, together with the fine-tuning comparison results against other methods.
>
> In summary, we hope the above response resolves your concerns and solidifies your support for our paper in the following discussion phases. Thanks again for your consideration.
>
> **References:**
>
> [1] FineWeb: Decanting the Web for the Finest Text Data at Scale. 2024
>
> [2] SimSMoE: Toward Efficient Training Mixture of Experts via Solving Representational Collapse. 2025
>
> [3] FLAME-MoE: A Transparent End-to-End Research Platform for Mixture-of-Experts Language Models. 2025

---

> > ### Author Rebuttal · Reviewer_S5JL · 2026-04-04
> >
> > Reviewer read the author rebuttal, which included explanations related to presentation and additional experiments (quantification of expert diversity, small-scale fine-tuning experiment). The authors should revise the paper accordingly. Most concerns are addressed.
> >
> > - What are the training datasets used for OLMoE vs. Flame-MoE? Describe the various datasets/tasks that are evaluated in Table 2 and new fine-tuning results (Alpaca dataset).
> >
> > - Fine-tuning: The experiment is conducted with a 98M-349M Flame-MoE. Results demonstrate relative benefits during training with MP-MoE. Only training loss is reported.
> >
> > - Clarify the content in Figure 2 in the paper.

---

> > > ### Author Response · Authors · 2026-04-04
> > >
> > > Thank you for the follow-up questions. Due to the character limit, we were unable to provide these details as clearly as we would have liked during the initial rebuttal phase. We provide a point-by-point response to your concerns below.
> > >
> > > **Q1: What are the training datasets used for OLMoE vs. Flame-MoE? Describe the various datasets/tasks that are evaluated in Table 2 and new fine-tuning results (Alpaca dataset).**
> > >
> > > For **Table 2**, both OLMoE and OLMoE + MP-MoE are trained on the same **FineWeb-EDU subset** under matched token budgets (1B, 5B, and 50B). FineWeb-Edu is the educational-content subset of FineWeb, which was introduced as a large-scale open pre-training corpus.
> > >
> > > The evaluation tasks in **Table 2** are **MMLU, BoolQ, HellaSwag, BBH, ARC-Easy, and ARC-Challenge**. These benchmarks cover complementary capabilities, including broad knowledge and reasoning (**MMLU**), yes/no reading comprehension (**BoolQ**), commonsense completion (**HellaSwag**), challenging multi-step reasoning (**BBH**), and grade-school science question answering (**ARC-Easy / ARC-Challenge**).
> > >
> > > The new fine-tuning result in the rebuttal uses a different backbone, namely **Flame-MoE**, and is intended as an additional transfer/generalization check beyond OLMoE. In this experiment, we first **pre-train Flame-MoE on FineWeb-EDU 10BT**, and then **fine-tune it on Alpaca**, comparing MoE, SimSMoE, and MP-MoE under the same setting. **Alpaca** is a widely used instruction-tuning dataset consisting of about 52K instruction-response examples, created using a modified Self-Instruct pipeline with text-davinci-003 for instruction-following fine-tuning. In this fine-tuning setting, **MP-MoE achieves the best final loss and perplexity (PPL)** among the compared methods.
> > >
> > > We will clarify the datasets and tasks more explicitly in the camera-ready version.
> > >
> > > **Q2: Fine-tuning: The experiment is conducted with a 98M-349M Flame-MoE. Results demonstrate relative benefits during training with MP-MoE. Only training loss is reported.**
> > >
> > > Thank you for raising this point. In our rebuttal fine-tuning experiments, we provided not only the **training loss** but also the **perplexity (PPL)**, which demonstrates the superior performance of MP-MoE compared to the other baselines. To further address this concern, we also evaluated the fine-tuned model on several small benchmark test sets, including **COPA** (commonsense causal reasoning), **RTE** (natural language inference), **OpenBookQA** (science question answering), **WSC** (pronoun resolution), and **HellaSwag** (contextual commonsense continuation). The corresponding results are shown below.
> > >
> > > | Benchmark | Pretrain | MoE fine-tuning | SimSMoE fine-tuning | MP-MoE fine-tuning |
> > > | :--- | :--- | :--- | :--- | :--- |
> > > | COPA | 0.5200 | 0.5700 | 0.5702 | **0.6200** |
> > > | HellaSwag | 0.2702 | 0.2706 | 0.2717 | **0.2835** |
> > > | OpenBookQA | 0.2600 | 0.2660 | 0.2660 | **0.2800** |
> > > | RTE | 0.5251 | 0.5255 | 0.5054 | **0.5487** |
> > > | WSC | 0.3554 | 0.3558 | 0.3462 | **0.5288** |
> > >
> > > We would like to emphasize that this fine-tuning study is intended as a **small-scale feasibility validation**. Even under this limited setting, **MP-MoE consistently shows improvements** over the baseline and competing method. Since the main focus of this paper is the effectiveness of MP-MoE during **pre-training**, we have not yet conducted large-scale fine-tuning or broader benchmark evaluations, which would require substantially more compute resources. We agree that this is an important direction, and we will further explore it in our future work.
> > >
> > > **Q3: Clarify the content in Figure 2 in the paper.**
> > >
> > > Figure 2 is meant to validate our core assumption that the expert co-occurrence matrix is a good proxy for expert similarity. Specifically, we compare, for the same MoE layer, (i) an **expert output-similarity matrix** computed from pairwise similarities of expert outputs on identical tokens, and (ii) a **co-occurrence matrix** computed from routing statistics.
> > >
> > > In **Figure 2(a)**, the **red points** are derived from the output-similarity matrix and the **blue points** are derived from the co-occurrence matrix; their close alignment in the t-SNE plots across Layers 2, 5, and 9 shows that the two matrices encode similar expert relationships. In **Figure 2(b)**, we further quantify this agreement using **CKA**, where the diagonal values (around 0.55–0.68) indicate a strong within-layer correspondence between output-based similarity and co-occurrence-based similarity.
> > >
> > > Therefore, the main message of Figure 2 is that **expert co-occurrence can serve as an efficient surrogate for inter-expert similarity**, which justifies our use of co-occurrence statistics to construct the covariance matrix in MP-MoE. We will clarify this more explicitly in the camera-ready version.
> > >
> > > In summary, we hope the above responses successfully address your additional concerns and further strengthen your support for our work.

---

### Decision · Program_Chairs · 2026-04-30

**Decision:**

Accept (regular)

**Comment:**

In this paper, the authors introduce MP-MoE (Mahalanobis-Pruned Mixture-of-Experts), a novel
routing framework that reimagines the expert selection process through the lens of ensemble pruning.
The core motivation is to address “representation collapse” and the “echo chamber” effect—where
top-k routing mechanisms repeatedly select highly correlated experts, thereby limiting the model’s
total capacity. By optimizing a Mahalanobis-distance-based objective, the authors explicitly encourage
expert diversity during inference. Reviewers generally agreed that the perspective of using the expert
co-occurrence matrix to model covariance is both original and technically elegant.

**Recommendation:** The strengths of the paper—specifically its novel ensemble-based perspective on routing and the effective mitigation of expert correlation—outweigh the minor concerns regarding performance margins. MP-MoE provides a fresh, theoretically grounded alternative to the heuristic-heavy routing strategies currently prevalent in the field. While Reviewer qQPC raised valid points regarding the theoretical depth of the pruning mechanism, these considerations are significantly outweighed by the paper’s primary contribution: a highly efficient, training-free framework that consistently delivers state-of-the-art compression-to-accuracy trade-offs across multiple large-scale architectures. Moreover, since Reviewer qQPC acknowledged that most of the concerns were addressed during the rebuttal. I therefore recommend acceptance, and I encourage the authors to include the expanded theoretical results provided during the rebuttal in the final version of the manuscript.